# GRL-SNAM: Geometric Reinforcement Learning with Differential Hamiltonians for Navigation and Mapping in Unknown Environments

**Aditya Sai Ellendula**[1]  **Yi Wang**[2]  **Minh Nguyen**[3]  **Chandrajit Bajaj**[1,2]
[1]Department of Computer Science, University of Texas at Austin
[2]Oden Institute, University of Texas at Austin
[3]Department of Mathematics, University of Texas at Austin
`{adityase,panzer.wy,minhpnguyen,bajaj}@utexas.edu`

## Abstract

We present GRL-SNAM, a geometric reinforcement learning framework for Simultaneous Navigation and Mapping in unknown environments. GRL-SNAM differs from traditional SLAM and other reinforcement learning methods by relying exclusively on local sensory observations without constructing a global map. Our approach formulates navigation and mapping as coupled dynamics on generalized Hamiltonian manifolds: sensory inputs are translated into local energy landscapes that encode reachability, obstacle barriers, and deformation constraints, while policies for sensing, planning, and reconfiguration evolve stagewise under Differential Policy Optimization. A reduced Hamiltonian serves as an adaptive score function, updating kinetic/potential terms, embedding barrier constraints, and continuously refining trajectories as new local information arrives. We evaluate GRL-SNAM on 2D deformable navigation tasks, where a hyperelastic robot learns to squeeze through narrow gaps, detour around obstacles, and generalize to unseen environments. We compare our method against *local reactive* baselines (PF, CBF, staged DWA), *global* A\* references (rigid, clearance-aware), and deep RL baselines (PPO, TRPO, SAC) under identical stagewise sensing constraints. GRL-SNAM achieves strong path quality while using the minimal map coverage, preserves clearance, and generalizes to unseen layouts. This demonstrates that our Hamiltonian-structured RL framework enables high-quality navigation through *minimal exploration* via local energy refinement rather than global mapping.

## 1 Introduction

Reinforcement learning has achieved strong performance in high-dimensional control, but continuous navigation in partially observed and changing environments remains challenging. Long horizons, multi-scale decisions, and online adaptation often make standard methods difficult to apply, especially in simultaneous navigation and mapping (SNAM), where an agent must move toward a goal while incrementally building and exploiting an evolving representation of the environment. In these settings, purely model-free methods can require many interactions, while hierarchical approaches often depend on manually designed decompositions that do not transfer well across environments.

A central difficulty is that most standard reinforcement learning (RL) policies do not explicitly encode the geometric and physical structure of navigation. As a result, they may rely heavily on statistical regularities in the training distribution rather than on invariant task structure, which can reduce robustness under distribution shift and degrade performance over long rollouts.

### 1.1 Beyond Standard Bellman Bootstrapping via Hamiltonian Updates

Our framework departs from the Bellman-recursive training loop that underlies most mainstream RL methods. Instead of learning a value function through recursive bootstrapping and then improving a policy from that estimate, our method parameterizes a Hamiltonian energy landscape whose gradients induce local control actions. The Hamiltonian parameters are learned directly from trajectory

feedback and stagewise adaptation signals, so training remains fully interactive and data-driven. At execution time, actions are computed in a feedforward manner from the current state, local observations, and the current Hamiltonian, avoiding rollout-based value propagation during control. While value-based methods and our dual formulations agree at optimality, our method explicitly learns in the non-optimal settings by modeling the dual Hamiltonian dynamics directly.

## 1.2 KEY INSIGHT: HAMILTONIAN STRUCTURE AS NAVIGATION INDUCTIVE BIAS

We propose to encode navigation through a learned Hamiltonian:

$$\mathcal{H}(q, p) = K(p) + P(q) \tag{1}$$

where kinetic and potential energies encode control objectives, constraints, and adaptation strategies. This formulation introduces three structural advantages:

(1) **Energy conservation** stabilizes long-horizon rollouts by preventing accumulation of numerical errors. (2) **Symplectic geometry** naturally separates fast reactive dynamics from slow strategic planning, addressing multi-scale temporal coordination. (3) **Barrier encoding** integrates safety and collision avoidance directly into potential functions, eliminating fragile reward shaping.

Hamiltonian structure transforms policy optimization into Differential Policy Optimization (Nguyen & Bajaj (2025)), where policies emerge as gradient flows of learned energies that respect geometry, conserve invariants, and generalize across environments. Because the Hamiltonian is a local encoding of a long-horizon objective, local gradient steps remain globally goal-directed through the shared energy landscape.

## 1.3 OFFLINE-ONLINE HAMILTONIAN SYNERGY

We distinguish between complementary learning regimes that exploit this geometric structure:

**Offline learning** fits reference Hamiltonians $h^{\theta^*}$ from interaction trajectories, capturing reusable geometric structure for sensing, path extraction, and reconfiguration in local frames. These learned reference energies serve as stable priors for downstream adaptation.

**Online adaptation** performs constrained, environment-dependent corrections to the reference Hamiltonian,

$$h^{\mathrm{adapted}} = h^{\mathrm{ref}} + \Delta h^{\mathrm{context}},$$

where $\Delta h^{\mathrm{context}}$ is inferred from the current obstacle configuration and task context. This yields conservative adaptation: the system preserves the learned reference geometry while adjusting only the energy components that are relevant to the new environment.

## 1.4 CONTRIBUTIONS

Our contributions include:

1. **A dual Hamiltonian RL formulation for SNAM**: We formulate navigation as learning a structured Hamiltonian energy landscape whose gradients induce local control actions, yielding a distinct dual alternative to standard value-function-based RL.

2. **Multi-scale geometric coordination**: Differential policies for sensing, planning, and adaptation unified through shared energy formulations, achieving temporal scale separation without manual hierarchy design.

3. **Physics-grounded adaptation**: Principled offline-online decomposition where stable reference dynamics adapt through geometric alignment rather than catastrophic relearning.

4. **Theoretical properties**: We analyze stability under temporal scale separation, structure preservation for the conservative Hamiltonian substeps, and additive sample-complexity scaling under independent module training.

5. **Empirical validation**: We validate the framework on deformable navigation tasks and compare against both generic deep RL baselines and task-specific navigation baselines.

## 2 RELATED WORK

We focus on structure-preserving, deployable navigation with deformable bodies. Our work lies at the intersection of geometric control, continuous-time reinforcement learning, safety-critical planning, and deformable robot navigation.

**Geometric Structure in RL and Control.** Most navigation-oriented RL methods operate in Euclidean spaces using standard PPO (Schulman et al. (2017)), SAC (Haarnoja et al. (2018)), or TD3 (Fujimoto et al. (2018)) objectives, without explicitly preserving geometric structure in the induced dynamics. Prior geometric approaches include SE(3)-equivariant policies (Hoang et al. (2025)) and Riemannian safe navigation (Klein et al. (2023)), which incorporate symmetry or manifold-aware projections. Hamiltonian and port-Hamiltonian neural models (Desai et al. (2021)) show that structure-preserving parameterizations can improve stability and generalization. However, these works focus on simpler control settings rather than modular, partially observed navigation.

**Continuous-Time RL.** Beyond standard dynamic-programming-based RL, recent work has explored continuous-time formulations that avoid or reinterpret Bellman recursion. Jia and Zhou (Jia & Zhou (2023)) derive continuous-time $q$-learning through a martingale characterization, Settai et al. (Settai et al. (2025)) study model-free temporal-difference learning for stochastic continuous dynamics through an HJB-oriented lens, and Nguyen and Bajaj (Nguyen & Bajaj (2025)) develop a dual Hamiltonian control perspective. Our work is most closely aligned with this emerging line, but differs in its focus on simultaneous navigation and mapping, modular Hamiltonian subpolicies, and stagewise online adaptation in geometry-rich environments.

**Safety-Critical Navigation.** Control Barrier Function (CBF) integration with RL achieves formal safety guarantees (Li et al. (2023)), but treats constraint satisfaction as orthogonal to navigation optimality, often resulting in conservative behaviors. Our Hamiltonian formulation integrates safety constraints directly within the energy structure.

**Deformable Robot Navigation.** Recent work demonstrates ring-like navigation through pre-programmed strategies: aerial gap navigation via fixed Liquid Crystal Elastomer responses (Qi et al. (2024)) and HAVEN (Mulvey & Nanayakkara (2024)) using predetermined shape-changing sequences. These approaches rely on offline parameter optimization followed by deterministic execution—they cannot adapt deformation strategies online as environmental conditions change.

**Neural Scene Representations.** NeRF-based SLAM methods like NICE-SLAM (Zhu et al. (2022)) provide rich environmental representations that complement our energy-based navigation formulation by supplying obstacle and free-space information for barrier and goal potential computation.

**Simultaneous Navigation and Mapping.** Most SNAM approaches prioritize building detailed maps before navigation. SGoLAM (Kim et al. (2021)) couples goal localization with occupancy mapping, CMP (Gupta et al. (2019)) integrates a differentiable planner into learned mapping, and CL-SLAM (Vödisch et al. (2023)) maintains maps for long-term adaptability. In contrast, our GRL-SNAM framework aims to *reach goals via high-quality, well-weighted paths while mapping as little of the environment as possible*. To our knowledge, no prior work explicitly targets minimal exploration; our method introduces progressive path refinement, continually improving least-cost trajectories as new observations arrive.

**Positioning.** GRL-SNAM extends Hamiltonian structure from low-dimensional control to simultaneous navigation and mapping with sensing, path extraction, and deformable-body reconfiguration. Unlike approaches that append safety filters to a learned controller or rely on pre-programmed deformation strategies, our method learns an energy landscape through interaction and executes via local Hamiltonian gradients. The modular architecture preserves specialization across subpolicies, while the shared Hamiltonian coupling provides a globally goal-directed objective and a principled mechanism for stagewise online adaptation.

## 3 METHODOLOGY

We present GRL-SNAM (Geometric Reinforcement Learning for Simultaneous Navigation and Mapping), a Hamiltonian-structured navigation framework that combines offline-learned subpolicies with online stagewise adaptation. The central object is a surrogate Hamiltonian defined on

phase space; its gradients generate local control actions for sensing, path extraction, and shape re-configuration, while a navigator adjusts environment-dependent weights to reflect current obstacle configurations.

We consider a deformable robot with state $q_t = (\boldsymbol{c}_t, \theta_t, \boldsymbol{y}_t, \psi_t) \in \mathcal{Q}$, where $(\boldsymbol{c}_t, \theta_t)$ denotes the robot pose in the world frame and $(\boldsymbol{y}_t, \psi_t)$ captures sensing and internal configuration variables. The robot moves from $\mathbf{x}_0$ to $\mathbf{x}_g$ in an unknown environment with binary occupancy map $I : \mathbb{R}^2 \to \{0, 1\}$. At each stage, the current local obstacle realization is summarized by an environment descriptor $\mathcal{E}$. GRL-SNAM operates in two coupled layers: offline, it learns three module-specific Hamiltonian response models; online, it assembles a surrogate Hamiltonian from these learned responses and performs stagewise corrections as $\mathcal{E}$ changes.

## 3.1 Online Stagewise Adaptation

We now describe how the Hamiltonian structure (§3.2), the modular policy architecture (§3.3), and the meta-learning framework (§C.4) are combined at test time into a unified navigation system.

**Overview.** At each navigation stage, the navigator sequentially queries the three offline-trained policies $\pi_y$ (sensor), $\pi_f$ (frame/FPE), and $\pi_o$ (object/reconfig) to obtain state-dependent control proposals. The meta-policy $g_\xi$ maps the current environment and policy responses to environment-dependent energy weights and friction, from which we assemble a surrogate Hamiltonian and integrate the corresponding port-Hamiltonian dynamics with dissipation and port correction. Observable quantities such as clearance, goal progress, and speed are then used to perform a short Jacobian-based update of the active energy weights and friction coefficients, yielding a stagewise adaptation loop. The full pseudo-code, including initialization, query protocols, energy assembly, integration details, and adaptation rules, is deferred to Algorithm 3 in Appendix E. Although the control law is evaluated locally at each step, the resulting action field is globally conditioned through the Hamiltonian weights, which depend on the sensed obstacle configuration and goal context.

## 3.2 Navigation as Hamiltonian Optimization

**From optimal control to Hamiltonian via Legendre–Fenchel conjugacy (fixed $\mathcal{E}$).** Fix an environment $\mathcal{E}$ and consider the control–affine dynamics $\dot{q} = f(q) + A(q)\,u$ with stage cost $L(q, u; \mathcal{E}) = -\mathcal{R}(q; \mathcal{E}) + \varphi(u)$, where $\mathcal{R}$ encodes goal/deflection/barrier terms and $\varphi$ penalizes effort. Pontryagin's principle introduces a costate $p$ and the *control Hamiltonian* $\mathcal{H}(q, p, u; \mathcal{E}) := p^\top(f(q) + A(q)u) - L(q, u; \mathcal{E})$. Eliminating $u$ amounts to taking the Legendre–Fenchel conjugate of $\varphi$:

$$H(q, p; \mathcal{E}) = \sup_u \left\{ p^\top A(q)u - \varphi(u) \right\} + p^\top f(q) + \mathcal{R}(q; \mathcal{E})$$
$$= \varphi^*\big(A(q)^\top p\big) + p^\top f(q) + \mathcal{R}(q; \mathcal{E}), \tag{2}$$

provided $\varphi$ is proper, closed, and strictly convex. The optimal feedback is $u^\star(q, p) = \nabla\varphi^*(A(q)^\top p)$. In the common quadratic case $\varphi(u) = \frac{1}{2}\,u^\top \Phi\,u$ (with $\Phi \succ 0$ as a kinetic "term"), we have $\varphi^*(w) = \frac{1}{2}\,w^\top \Phi^{-1}w$, hence

$$H(q, p; \mathcal{E}) = \tfrac{1}{2}\,p^\top\big(A(q)\Phi^{-1}A(q)^\top\big)p + p^\top f(q) + \mathcal{R}(q; \mathcal{E}). \tag{3}$$

Identifying the *inverse mass* as $M(q)^{-1} := A(q)\Phi^{-1}A(q)^\top$ and (optionally) absorbing $p^\top f(q)$ into a gauge term (or set $f(q) \equiv 0$) yields the mechanical form $H(q, p; \mathcal{E}) = \frac{1}{2}\,p^\top M(q)^{-1}p + \mathcal{R}(q; \mathcal{E})$. The canonical equations, $\dot{q} = \nabla_p H$ and $\dot{p} = -\nabla_q H$, are therefore the *Hamiltonian outcome* of the fixed-$\mathcal{E}$ optimal control problem. Soft constraints (barriers) simply contribute additively to $\mathcal{R}$; nonconservative effects can be modeled as port inputs without altering the conjugate construction (esp. friction). Thus, for each scenario $\mathcal{E}$, the inner motion law is Hamiltonian with kinetic energy induced by the control penalty via conjugacy and potential shaped by the environment. Note that motion planning offline policy may or may not follow the surrogate Hamiltonian one wish to align but we calibrate the surrogate by interaction with response.

**Search space for the Hamiltonian.** The goal of navigator is to **learn to search** in the energy space of $H$. The Hamiltonian defined in 3 is a function on the cotangent bundle $T^*\mathcal{Q}$:

$$H \in \mathcal{H} := \big\{ H(q, p; \mathcal{E}) = \tfrac{1}{2}\,p^\top M(q)^{-1}p + \mathcal{R}(q; \mathcal{E}) \,\big|\, M : \mathcal{Q} \to \mathbb{S}^2_{++},\ \mathcal{R} \in \mathscr{R} \big\}.$$

We regard $\mathscr{R}$ as a Hilbert space of admissible potentials on $\mathcal{Q}$ (and w.r.t. environmental configuration) (e.g. $L^2(\mathcal{Q} \times \mathscr{E})$). For planar navigation we *restrict* the search to the environment-indexed linear cone generated by task energies, where each policy governs a distinct energy term and a joint dynamical barrier term. We model the meta navigator by a parametrized map $\mathcal{E} \to \eta_\xi(\mathcal{E})$ producing nonnegative dual weights that shape the primal potential. In general, each energy term may itself be parametrized:

$$H(q, p; \omega, \xi, \mathcal{E}) \;=\; \tfrac{1}{2} p^\top M(q; \omega_M)^{-1} p \;+\; \mathcal{R}(q; \omega, \eta_\xi(\mathcal{E})),$$

$$\mathcal{R}(q; \omega, \eta_\xi(\mathcal{E})) \;=\; E_{\text{sensor}}(q; \mathcal{E}, \omega_y) + \beta(\mathcal{E}) \, E_{\text{goal}}(q; \mathcal{E}, \omega_g) + \lambda(\mathcal{E}) \, E_{\text{obj}}(q; \omega_d)$$

$$+ \sum_{i \in \mathcal{C}_t(\mathcal{E}, q)} \alpha_i(\mathcal{E}, t) \, b\big(d_i(q; \mathcal{E}); \omega_b\big), \tag{4}$$

with $\eta_\xi(\mathcal{E}) = (\beta(\mathcal{E}), \lambda_{\text{obj}}(\mathcal{E}), \{\alpha_i(\mathcal{E}, t)\}) \in \mathbb{R}_+^{m(\mathcal{E}, t)}$. Here $\omega = (\omega_y, \omega_M, \omega_g, \omega_d, \omega_b)$ are *intra-term* parameters (e.g. metric, goal shape, deformation model, barrier template), while $\eta_\xi$ learns the *inter-term* tradeoffs by mapping the environment $\mathcal{E}$ to dual weights. The cardinality $m(\mathcal{E}, t) = 2 + |\mathcal{C}_t(\mathcal{E}, q)|$ is environment/active-set dependent, so $\eta_\xi$ is implemented with a permutation-invariant set encoder that outputs per-constraint scores $\alpha_i(\mathcal{E}, t) \geq 0$, together with scalars $\beta(\mathcal{E}), \lambda(\mathcal{E}) \geq 0$. The active set $\mathcal{C}_t(\mathcal{E}, q) := \{i \mid d_i(q, \mathcal{E}) \leq \hat{d}\}$ is discovered online by sensing.

## 3.3 Navigator's SubModular Architecture

Rather than learning a single navigation policy, we decompose the problem into three independent score functions, each dedicated to a specific navigation aspect (see Figure 1 for details):

**Definition 3.1** (Independent Score Functions). *Let $\mathcal{K} = \{y, f, o\}$ denote the set of policy indices corresponding to sensor, frame, and object domains respectively. For each $k \in \mathcal{K}$, define:*

- *$z_k \in \mathcal{Z}_k$: the phase space state for policy $k$, where $\mathcal{Z}_k = \mathcal{Q}_k \times \mathcal{P}_k$ with configuration space $\mathcal{Q}_k$ and momentum space $\mathcal{P}_k$*

- *$\theta_k \in \Theta_k$: the learnable parameters for policy $k$, where parameter sets satisfy disjointness: $\Theta_i \cap \Theta_j = \emptyset$ for $i \neq j$*

- *$h_k^{\theta_k} : \mathcal{Z}_k \times \mathscr{E} \times \mathbb{R}_{\geq 0} \to \mathbb{R}$: a learned energy functional parameterized by $\theta_k$*

*Each policy $\pi_k$ is defined as an independent score function: $s_k^{\theta_k}(z_k, \mathcal{E}, t) = S_k^{\theta_k}(\nabla_{z_k} h_k^{\theta_k}(z_k, \mathcal{E}, t))$*

*The parameter disjointness ensures independence: $\frac{\partial s_k^{\theta_k}}{\partial \theta_j} = 0$ for all $j \neq k$, allowing parallel training while maintaining coordination through shared constraints $\mathcal{C}_t$.*

**Policy Abstraction.** Each policy is an offline-learned module exposed through a standardized query-response interface:

- **Sensor Policy** ($\pi_y$): Adapts perception parameters $\to$ energy gradients for information gathering
- **Frame Policy** ($\pi_f$): Plans collision-free paths $\to$ energy gradients for goal attraction
- **Shape Policy** ($\pi_o$): Controls robot deformation $\to$ energy gradients for obstacle navigation

The key insight is that our Navigator is agnostic to policy implementation—our contribution is the Hamiltonian structure binding them together through dynamic constraint sets $\mathcal{C}_t$.

Algorithm 3 details the online adaptation procedure, where the navigator issues sequential queries to the sensor, frame, and reconfig policies, integrates their energy gradients into a Hamiltonian update, and applies meta-corrections for contextual alignment to generate stable trajectories for new environments.

**Hamiltonian of modular sub-systems.** Let $\mathcal{K} = \{y, f, o\}$ index three Hamiltonian submodules with local states $z_k = (q_k, p_k) \in T^* \mathcal{Q}_k$ and local Hamiltonians

$$H_k(q_k, p_k; \xi, \mathcal{E}) = \tfrac{1}{2} p_k^\top M_k(q_k)^{-1} p_k \;+\; \underbrace{\mathcal{R}_k(q_k; \mathcal{E})}_{\text{module potential}} .$$

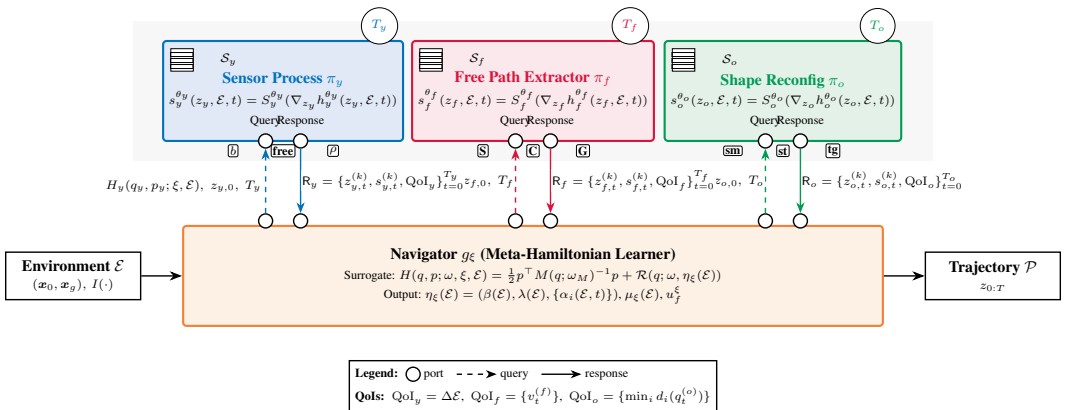

Figure 1: Independent score function architecture and query–response interface. The Navigator $g_\xi$ issues queries containing local Hamiltonians $H_k$, initial states $z_{k,0}$, and time horizons $T_k$ to each policy $\pi_k$ ($k \in \{y, f, o\}$). Each policy computes score functions $s_k^{\theta_k}$ via energy gradients from learned Hamiltonians $h_k^{\theta_k}$, backed by spatial indices $\mathcal{S}_k$ for efficient neighbor queries. Policies return standardized responses $\mathsf{R}_k$ containing state trajectories, score sequences, and QoIs. The Navigator aggregates these to update the surrogate Hamiltonian and generate meta-corrections $\eta_\xi(\mathcal{E})$, $\mu_\xi(\mathcal{E})$, and $u_f^\xi$ for adaptive navigation.

The *navigator* represents the whole stack by a *surrogate* Hamiltonian on $T^*\mathcal{Q}$, where $\eta(\mathcal{E}) = g_\xi(\mathcal{E}) \in \mathbb{R}_+^{m(\mathcal{E},t)}$ is the meta policy(explorer) output for potential functional. For navigation, in particular the reference potential for each submodule is:

$$\mathcal{R}_y(q_y; \xi, \mathcal{E}) = E_{\text{sensor}}(q_y; \mathcal{E}, \omega_y) + \sum_{i \in \mathcal{C}_t(\mathcal{E}, q)} \alpha_i(\mathcal{E}, t)\, b\big(d_i(q; \mathcal{E}); \omega_b\big) \tag{5}$$

$$\mathcal{R}_f(q_f; \xi, \mathcal{E}) = \beta(\mathcal{E})\, E_{\text{goal}}(q_f; \mathcal{E}, \omega_g) + \sum_{i \in \mathcal{C}_t(\mathcal{E}, q)} \alpha_i(\mathcal{E}, t)\, b\big(d_i(q; \mathcal{E}); \omega_b\big) \tag{6}$$

$$\mathcal{R}_o(q_o; \xi, \mathcal{E}) = \lambda(\mathcal{E})\, E_{\text{obj}}(q_o; \omega_d) + \sum_{i \in \mathcal{C}_t(\mathcal{E}, q)} \alpha_i(\mathcal{E}, t)\, b\big(d_i(q; \mathcal{E}); \omega_b\big) \tag{7}$$

**Remark.** *A black-box setup assume one cannot observe the potential components $\{\mathcal{R}_k\}$ yet can only observe kinectic terms by the integrated dynamics via policy $\pi_k$. A gray-box setup can allow navigator reshape potential (and possibly kinectics) so one do not need to caliberate surrogate energy (equation 3) with local Hamiltonian aggregation $\hat{H} := \sum_{k \in \mathcal{K}} H_k^{\theta_k}$.*

**Hamiltonian Dynamics of modular sub-systems.** Each submodule $k \in \mathcal{K}$ integrates its local dynamics for a short horizon and returns a standardized response $\mathsf{R}_k$. Let the *effective module Hamiltonian* be $h_k^{\theta_k}(q_k, p_k, t)$ with initial condition $h_k^{\theta_k}(q_k, p_k, 0) = H_k(q_k, p_k; \mathcal{E})$. The local (port-)Hamiltonian flow with dissipation and remaining nonconservative input is

$$\dot{q}_k = \nabla_{p_k} h_k^{\theta_k}(q_k, p_k, t), \tag{8}$$

$$\dot{p}_k = -\nabla_{q_k} h_k^{\theta_k}(q_k, p_k, t) - \Gamma_k^\xi(q_k; \mathcal{E})\, \nabla_{p_k} h_k^{\theta_k}(q_k, p_k, t) + G_k^\xi(q_k; \mathcal{E})\, u_k^\xi(q_k, p_k, t, \mathcal{E}), \tag{9}$$

with $\Gamma_k \succeq 0$ a Rayleigh/viscous damping and $G_k^\xi u_k^\xi$ the *nonconservative* (non-potential) external input. We define the *score of dynamics* recorded by module $k$ as the deterministic drift,

$$s_k(z_k, t) := \begin{bmatrix} \nabla_{p_k} h_k^{\theta_k} \\ -\nabla_{q_k} h_k^{\theta_k} \end{bmatrix} + \underbrace{\begin{bmatrix} 0 \\ -\Gamma_k^\xi \nabla_{p_k} h_k^{\theta_k} + G_k^\xi u_k^\xi \end{bmatrix}}_{\text{non-Hamiltonian contributions (friction/ports)}}, \qquad z_k = (q_k, p_k).$$

**Navigator meta-learning details.** For clarity of exposition, we defer the full formulation of the navigator as a meta-Hamiltonian learner including the construction of $\mathcal{R}(q; \eta_\xi(\mathcal{E}))$, the training objective in equation 48, and the QoI-based online adaptation scheme to Appendix C.4.

### 3.4 MULTI-SCALE TEMPORAL COORDINATION

The policies operate at natural temporal hierarchies, creating stable multi-scale coordination:

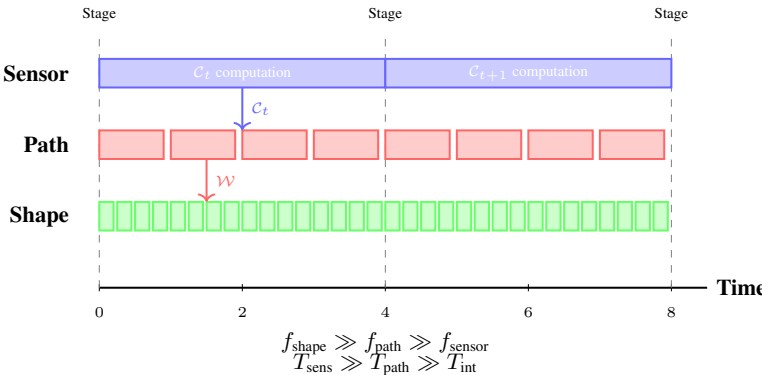

Figure 2: Temporal hierarchy. Sensor policy operates at low frequency (once per stage), establishing environmental constraints $\mathcal{C}_t$. Path policy operates at medium frequency within each stage, computing waypoints $\mathcal{W}$. Shape policy operates at high frequency, continuously adapting at each integration step. This creates a natural hierarchy where slow sensor updates provide stable constraints for faster path and shape adaptations.

This temporal separation enables a **nested quasi-static approximation**: the fastest dynamics (reconfiguration) equilibrate within each frame update, and frame dynamics settle before the slower sensor policy evolves. This hierarchy prevents destabilizing interactions across timescales while preserving the necessary coupling for coherent, coordinated behavior.

### 3.5 OFFLINE PHYSICS LEARNING VS ONLINE ADAPTIVE CORRECTION

Our approach resolves the fundamental tension between learning complex dynamics and real-time adaptation through principled decomposition:

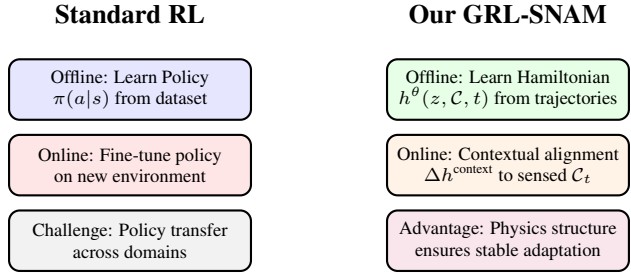

Figure 3: Comparison between standard RL offline/online adaptation and our physics-grounded approach. Standard methods learn arbitrary policies and struggle with transfer, while our approach learns physically meaningful Hamiltonians that naturally adapt to environmental variations.

### 3.6 THEORETICAL PROPERTIES

Our framework provides three key theoretical guarantees:

**Theorem 3.2** (Multi-Policy Stability). *Under temporal scale separation $T_{sens} \gg T_{path} \gg T_{int}$ and bounded parameter updates, the coupled system maintains stability with error bound $\mathcal{E}_{total} \le \epsilon$.*

**Theorem 3.3** (Symplectic Preservation). *Each score function generates symplectic dynamics preserving the canonical structure $\omega_k(z_{k,t+1}) = \omega_k(z_{k,t})$.*

**Theorem 3.4** (Linear Sample Complexity). *Independent training achieves total sample complexity $N_{total} = \sum_{k \in \{y,f,o\}} O(\epsilon_k^{-(2d_k+4)})$, linear in the sum of policy dimensions rather than exponential in joint dimensionality.*

We defer the proof of theorems in appendix. The system thinks in physics during offline training but adapts through energy corrections during online execution, combining principled dynamics stability with real-world deployment flexibility.

## 4 EXPERIMENTS

We evaluate GRL-SNAM across multiple dimensions that highlight the unique capabilities of our geometric approach compared to standard reinforcement learning and classical navigation methods. Our evaluation encompasses task performance, safety guarantees, and learning efficiency under minimal sensing constraints. For more detailed results and analysis, refer to Appendix H

**Experimental Setup.** We evaluate GRL-SNAM in procedurally generated 2D deformable navigation tasks, where a hyperelastic ring must traverse cluttered environments with narrow gaps and varying obstacle densities. The robot perceives only a local window of size $2\hat{d} \times 2\hat{d}$, from which we construct a Hamiltonian energy functional with goal-directed potential $F_g$, barrier potentials $F_{bs}$, and adaptive coefficients $(\beta, \gamma, \alpha)$ modulated by context encoders.

**Baselines.** We compare GRL-SNAM against three classes of baselines under matched information constraints. In the **global planning** category, we consider rigid A* with obstacle inflation and deformable A* with a clearance-aware penalty. In the **local reactive** category, we include Potential Fields (PF), Control Barrier Functions (CBF), and a staged Dynamic Window Approach (DWA), all using the same local sensing window and stage management as GRL-SNAM. We also include **learning-based deep RL baselines**, PPO, TRPO, and SAC, trained with the same observation space and short-rollout setting as our method. This setup enables a fair comparison against both task-specific navigation pipelines and standard RL methods.

**Metrics.** Success Rate, Success-weighted Path Length (SPL), Detour Ratio, Minimum Clearance, Path Smoothness, Collisions, and Mapping Ratio (fraction of environment observed).

**Codebase.** Our implementation is available at: https://github.com/CVC-Lab/GRL-SNAM

### 4.1 MAIN RESULTS

**Q1. How efficiently does GRL-SNAM trade mapping for navigation quality?** Table 5 demonstrates that GRL-SNAM achieves CBF-level navigation quality (SPL = 0.95, Detour = 1.09) while using essentially the same minimal map coverage as PF (10.7% vs. CBF's 11.2%). This validates that our stagewise Hamiltonian refinement extracts maximum value per sensed environment unit.

For each deep RL algorithm (PPO, TRPO, SAC) we train three control parameterizations under the *same* short-rollout distribution and local observation space as GRL-SNAM: (i) a kinematic controller (policy outputs velocities), (ii) a dynamic controller (policy outputs forces integrated by a damped point-mass model), and (iii) a coefficient controller (policy outputs the Hamiltonian force-field coefficients $(\alpha, \beta, \gamma)$ used by GRL-SNAM). The aggregated PPO/TRPO/SAC rows in Table 5 summarize the best-performing configuration for each family. The best TRPO/SAC results reach at most SPL = 0.57 with almost grazing clearances (MinClear $\approx$ 0) and require larger mapping ratios ($\approx$ 14–15%), while PPO collapses to SPL = 0.07 with negative effective clearance. In contrast, GRL-SNAM attains high SPL and positive clearance under a strictly smaller sensing budget.

**Q2. Does GRL-SNAM outperform classical, reactive, and RL planners in complex environments?** Yes. Figure 9 shows GRL-SNAM achieves near-perfect success rates ($\approx 100\%$) across both in-distribution and out-of-distribution test cases, while all baselines degrade significantly. GRL-SNAM consistently maintains high SPL ($\approx 1.0$) with low variance and produces the smoothest trajectories with lowest turning angles. The Pareto frontier analysis confirms GRL-SNAM uniquely dominates the safety-performance trade-off.

Table 1: Navigation quality comparison (success-only runs). GRL-SNAM achieves near-CBF efficiency with minimal mapping budget, while deep RL baselines trained on the same short-rollout distribution and local observations yield lower SPL, larger detours, and smaller clearances.

| Method | SPL ↑ | Detour ↓ | Min. Clearance (m) ↑ | Mapping Ratio (%) ↓ |
|--------|-------|----------|----------------------|---------------------|
| PF | 0.77 | 1.42 | 0.18 | 10.3 |
| CBF | 0.96 | 1.04 | **0.32** | 11.2 |
| GRL-SNAM | **0.95** | **1.09** | 0.26 | **10.7** |
| PPO | 0.07 | 1.65 | -0.09 | 14.7 |
| TRPO | 0.57 | 1.44 | 0.004 | 14.3 |
| SAC | 0.57 | 1.53 | 0.004 | 14.6 |

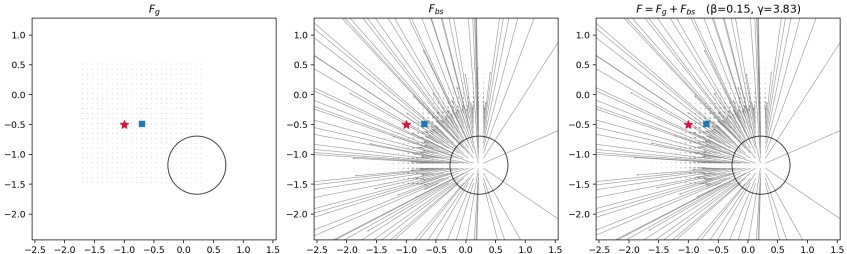

Figure 4: Hamiltonian force field composition. Left: goal force $F_g$; Middle: barrier forces $F_{bs}$; Right: adaptive combination yielding safe, goal-directed trajectories.

**Q3. How does the Hamiltonian formulation enable coherent navigation?** Figure 4 illustrates how GRL-SNAM unifies goal attraction $F_g$ and barrier repulsion $F_{bs}$ into a coherent navigation field through adaptive coefficients. Unlike reactive methods that treat forces independently, our differential composition $F = \beta F_g + \gamma F_{bs}$ creates contextually balanced dynamics that simultaneously pursue goals and avoid obstacles, which aligns with the high SPL and positive clearances observed in Table 5.

**Q4. What distinguishes GRL-SNAM's online adaptation from standard RL approaches?** Unlike standard deep RL policies that learn a fixed mapping from observations to actions (or coefficients), GRL-SNAM modifies the *entire local energy landscape* as new obstacles are sensed. Figure 10 demonstrates that the coefficients $(\beta, \gamma, \alpha)$ evolve dynamically to redefine the reduced Hamiltonian itself, ensuring energy-consistent posterior updates rather than heuristic reactive adjustments. This online reshaping of the Hamiltonian explains why GRL-SNAM maintains high success, SPL, and clearance under the same sensing budget where PPO/TRPO/SAC (in all three control variants) either collide, stall, or take inefficient detours.

**Deep RL baselines.** To contextualize the algorithmic contribution of GRL-SNAM, we additionally evaluate strong deep RL baselines (PPO, TRPO, and SAC) implemented under the *same* sensing pipeline, observation structure, action space, and Transformer encoder as our method. For fairness, all baselines operate on the identical short-rollout stagewise dataset derived from the dungeon environment of Liang et al. (2023), with horizon $H \in [2, 6]$, stage-exit goals, and locally reconstructed obstacles. Each agent outputs continuous 2D velocity actions and is trained with identical shaped rewards (goal progress, smoothness, and terminal imitation).

Table 2 summarizes performance under these matched conditions: despite millions of interaction steps, PPO/TRPO/SAC achieve only 18–26% success, whereas GRL-SNAM attains 87.5% with an order-of-magnitude fewer updates. This highlights that, even under identical data, sensing, and architectures, GRL-SNAM's Hamiltonian force-learning offers a substantially more reliable and sample-efficient mechanism for local navigation.

Table 2: Short-rollout navigation performance under identical sensing, rollouts, and architecture.

| Method | Success (%) ↑ | Mean State Error (m) ↓ | Mean Goal Dist. (m) ↓ |
|---|---|---|---|
| PPO | 26.1 | 1.8 | 1.2 |
| TRPO | 21.7 | 2.1 | 1.5 |
| SAC | 18.4 | 2.4 | 1.9 |
| **GRL-SNAM** | **87.5** | **0.3** | **0.1** |

## 4.2 ABLATION AND ROBUSTNESS STUDY

We conducted comprehensive ablation studies on loss components ($\mathcal{L}_{\text{friction}}$, $\mathcal{L}_{\text{multi}}$) confirming that friction matching is critical for stability while multi-start robustness prevents over-conservatism. Robustness evaluations under sensor noise and dynamics perturbations show significant degradation (87% success under severe noise vs 99% nominal) due to adaptive Hamiltonian framework. Sample analysis demonstrates faster convergence than RL baselines due to physics-informed structure. Refer to Appendix H for more details.

## 4.3 NAVIGATION IN COMPLEX MAP LAYOUTS

To test our method in more complex map layouts, we also evaluate GRL-SNAM on a point-agent navigation task in dungeon-style environments from Liang et al. (2023). The agent is a continuous 2D point with action $\mathbf{a}_t = [v_x, v_y] \in [-3,3]^2$ and receives only local observations. At each stage, a `StagewiseSensor` provides the current macro-cell exit as the local goal, while an `ObstacleExtractor` extracts visible obstacle primitives inside the sensing window $\mathcal{W}(q_t)$. GRL-SNAM uses a quadratic goal energy and radial obstacle barriers, and is compared against PPO, TRPO, and SAC under the same stagewise observation and action interface.

Table 3: Short-rollout baselines vs. GRL-SNAM on the point-agent navigation task.

| Algorithm | Success (%) | Mean state error (m) ↓ | Mean goal dist. (m) ↓ | Env. training steps |
|---|---|---|---|---|
| PPO | 26.1 | 1.8 | 1.2 | ≈ 3.2M env. steps |
| TRPO | 21.7 | 2.1 | 1.5 | ≈ 3.8M env. steps |
| SAC | 18.4 | 2.4 | 1.9 | ≈ 4.1M env. steps |
| GRL-SNAM | **87.5** | **0.3** | **0.1** | 500k gradient steps |

The RL baselines remain less reliable and less accurate than GRL-SNAM. Under the same local sensing and action interface, the Hamiltonian-structured surrogate achieves markedly higher success with lower state and goal error. This suggests that the gain comes primarily from the learning formulation rather than from the observation interface or action parameterization. Additional details are provided in Appendix H.

## 5 CONCLUSION

We introduced GRL-SNAM, a reinforcement learning framework that leverages Hamiltonian structure to couple sensing, planning, and deformation into a unified energy-based policy. Our formulation enables stable, feedforward navigation updates and achieves near-optimal path quality with minimal mapping effort in challenging deformable-robot tasks. The results highlight that incorporating geometric priors into RL can yield both efficiency and robustness, even under noisy sensing and out-of-distribution layouts. Future work will extend the approach to richer sensing modalities and more complex environments, with the goal of validating its scalability to real robotic systems.

### ACKNOWLEDGMENTS

This research was supported in part from the Peter O'Donnell Foundation, the Jim Holland-Backcountry Foundation and in part from a grant from the Army Research Office accomplished under Cooperative Agreement Number W911NF-19-2-0333.

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

## A    EXTENDED INTRODUCTION AND MOTIVATION

This section provides expanded context for the challenges addressed by GRL-SNAM and detailed justification for our geometric approach.

### A.1    COMPREHENSIVE ANALYSIS OF RL LIMITATIONS IN NAVIGATION

Contemporary reinforcement learning methods face several critical limitations that become particularly pronounced in continuous navigation tasks:

**Sample Efficiency Bottlenecks.** Standard RL algorithms like SAC (Haarnoja et al. (2018)) and PPO (Schulman et al. (2017)) require millions of environment interactions to learn effective navigation policies. This inefficiency stems from the curse of dimensionality in continuous control settings where the action space is infinite-dimensional and policies must simultaneously master fine-grained motor control and high-level strategic reasoning. In real-world deployment scenarios where data collection is expensive, this sample complexity becomes prohibitive.

The problem is exacerbated by the need for exploration in high-dimensional spaces. Unlike discrete control problems where systematic exploration strategies like $\epsilon$-greedy or UCB can provide theoretical guarantees, continuous control requires sophisticated exploration mechanisms that often rely on injected noise or entropy bonuses. These mechanisms frequently lead to unsafe or inefficient exploration behaviors that are unsuitable for real-world navigation tasks.

**Generalization Failures.** Policies trained in specific environments exhibit catastrophic performance degradation when deployed in novel settings, even when new environments share similar structure. This brittleness stems from the lack of inductive bias in standard neural network architectures. Without explicit encoding of physical principles or geometric structure, learned policies tend to memorize environment-specific features rather than discovering generalizable navigation principles.

The generalization problem is particularly acute in navigation because environmental variations can affect multiple aspects of the task simultaneously: obstacle configurations change collision constraints, surface properties affect dynamics, and lighting conditions influence perception. Standard RL approaches learn monolithic mappings that cannot decompose these variations into their constituent factors, leading to behaviors that fail when any component deviates from training conditions.

**Temporal Decomposition Challenges.** Navigation inherently requires coordination across multiple timescales: immediate obstacle avoidance operates on millisecond timescales, local path planning unfolds over seconds, and strategic goal-directed behavior spans minutes or hours. Standard RL algorithms struggle to learn policies that reason effectively across these scales, often getting trapped in locally optimal behaviors that satisfy short-term objectives while failing to make long-term progress.

Existing approaches to multi-scale reasoning such as hierarchical RL (Sutton et al. (1998)), options frameworks (Precup (2000)), or feudal networks (Vezhnevets et al. (2017)), typically require manual decomposition of the task space and careful engineering of reward functions for different levels. These methods introduce additional complexity without fundamentally addressing the structural issues that make multi-scale learning difficult.

### A.2    THE SNAM CHALLENGE: WHY STRUCTURE MATTERS

Simultaneous Navigation and Mapping (SNAM) represents a particularly challenging instance of the navigation problem where agents must build environmental representations online while traversing unknown spaces. This challenge amplifies limitations of RL approaches in several ways:

**Memory and Representation Learning.** SNAM requires policies to maintain and update spatial representations based on sensory observations. This places enormous demands on the policy's memory architecture, requiring it to simultaneously master memory management, spatial reasoning, and motor control. Standard recurrent architectures like LSTMs or GRUs struggle with this multifaceted learning problem, often failing to maintain coherent spatial representations over long episodes.

**Exploration-Exploitation Tradeoffs.** In SNAM, exploration serves dual purposes: gathering information about the environment for mapping and discovering navigation strategies. This creates complex exploration-exploitation tradeoffs that standard RL exploration mechanisms cannot handle

effectively. Random exploration may discover new regions but fails to systematically map environmental structure, while directed exploration based on current maps may miss critical features.

**Dynamic Environmental Coupling.** Unlike traditional navigation where environments are static, SNAM requires reasoning about how the agent's actions affect both its position and its knowledge of the environment. This creates a coupled learning problem where navigation decisions influence future mapping accuracy, and mapping quality affects navigation performance. Standard RL frameworks treat these as separate problems, missing the critical coupling that enables efficient SNAM.

Recent approaches in simultaneous navigation and mapping (SNAM) have coupled local mapping with policy learning to improve navigation performance. For example, SGoLAM (Kim et al. (2021)) interleaves goal localization with occupancy mapping to enable point-goal navigation, while Cognitive Mapping and Planning (CMP) (Gupta et al. (2019)) integrates a differentiable planner into a learned mapping framework. Continual SLAM (CL-SLAM) (Vödisch et al. (2023)) further emphasizes long-term adaptability by maintaining and updating maps during navigation. However, these methods rely on progressively constructing detailed maps of the environment before exploiting them for navigation. In contrast, our objective is to *reach the goal along high-quality, well-weighted paths while mapping as little of the unknown environment as possible*. To the best of our knowledge, no prior work explicitly formulates navigation with minimal exploration as the central goal. Our proposed GRL-SNAM framework achieves this by progressively refining paths: from observed environmental variations, the policy differentially learns to identify the least-cost trajectory, such that the path improves continuously as new local information is revealed.

### A.3 GEOMETRIC STRUCTURE: THE INEVITABLE SOLUTION

The limitations outlined above are not merely implementation details but fundamental consequences of treating navigation as unstructured optimization. Several lines of evidence suggest that geometric structure is not just helpful but inevitable for solving complex navigation problems:

**Physical Realizability.** Real robotic systems operate under physical constraints imposed by conservation laws, kinematic limitations, and actuator dynamics. Policies that violate these constraints cannot be implemented on physical systems, yet standard RL approaches have no mechanism to enforce such constraints during learning. Geometric formulations naturally incorporate physical constraints through the mathematical structure of the problem.

**Stability Requirements.** Long-horizon navigation requires numerical stability over extended rollouts. Standard neural network policies accumulate errors over time, leading to unstable behaviors in long episodes. Hamiltonian formulations with symplectic structure preserve important invariants (energy, momentum) that ensure stability over arbitrarily long rollouts.

**Compositionality Needs.** Complex navigation tasks require composing simpler behaviors: obstacle avoidance, path following, goal seeking, and environmental adaptation. Standard RL approaches learn monolithic policies that cannot decompose into interpretable components. Geometric formulations enable natural decomposition through energy terms that can be composed, weighted, and adapted independently.

### A.4 DIFFERENTIAL POLICY OPTIMIZATION: BEYOND FIXED POLICIES

Traditional RL optimizes fixed policy parameters $\theta$ to maximize expected returns over discrete timesteps. Differential Policy Optimization (Nguyen & Bajaj (2025)) fundamentally reconceptualizes this by learning dynamics operators through a continuous-time differential dual formulation.

**Mathematical Foundation.** Rather than directly learning policies, Differential Policy Optimization reformulates RL through continuous-time optimal control. By approximating discrete reward sums with time integrals:

$$\max_{\pi} \mathbb{E}\left[\sum_{k=0}^{H-1} r(s_k, a_k)\right] \approx \max_{\pi} \mathbb{E}\left[\int_0^T r(s_t, a_t)dt\right] \tag{10}$$

Applying Pontryagin's Maximum Principle introduces adjoint variables $p$ and defines the Hamiltonian function:

$$\mathcal{H}(p, s, a) := p^T f(s, a) - r(s, a) \tag{11}$$

The key insight is that optimal actions can be implicitly represented through the stationarity condition $\frac{\partial \mathcal{H}}{\partial a} = 0$, yielding the reduced Hamiltonian:

$$\hbar(s, p) := \mathcal{H}(s, p, a^*(s, p)) \tag{12}$$

**Score Function Learning.** Differential Policy Optimization learns a score function $g(x) \approx \hbar(x)$ where $x = (s, p)$ combines state and adjoint variables. The dynamics operator is constructed as:

$$G(x) = x + \Delta S \nabla g(x) \tag{13}$$

where $S = \begin{bmatrix} 0 & I \\ -I & 0 \end{bmatrix}$ is the canonical symplectic matrix and $\Delta$ is the discretization step.

**Stagewise Learning Advantages.** Unlike methods requiring backward-in-time adjoint calculations (as in Pontryagin's Maximum Principle), Differential Policy Optimization enables feedforward learning where each stage $t$ defines a local Hamiltonian $\mathcal{H}_t$ integrated forward in time:

$$\theta_{t+1} = \theta_t - \eta \nabla_\theta \mathcal{H}_t \tag{14}$$

This avoids the computational complexity and numerical instability of adjoint methods while maintaining theoretical guarantees through the geometric structure of the Hamiltonian formulation.

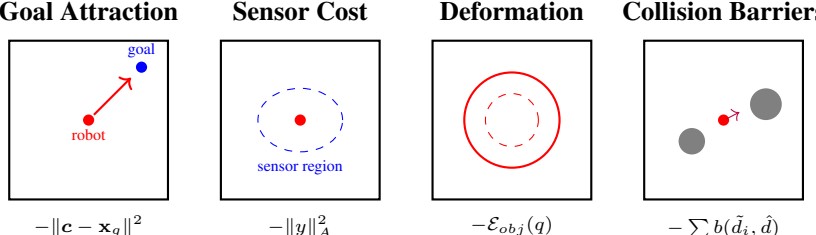

Figure 5: Policy-aligned energy decomposition. Each policy governs a distinct energy component: the Sensor Policy minimizes sensor cost, the FPE balances goal attraction and collision barriers, and the Reconfig Policy adapts size through deformation energy. Together these terms define the Hamiltonian reward $\mathcal{R}$.

## A.5 MULTI-POLICY ARCHITECTURE DETAILS

Our multi-policy decomposition addresses temporal scale separation through three specialized components operating at different timescales:

**Sensor Policy ($\pi_y$):** Operates at slow timescales to adapt perception strategies based on stagewise environmental feedback. This policy learns to focus attention on relevant environmental features, adjust sensor parameters for optimal information gain, and filter sensory noise. The sensor policy outputs constraints $\mathcal{C}_t$ that inform slower planning processes.

**Frame Policy ($\pi_f$):** Operates at medium timescales to plan collision-free trajectories in local coordinate frames. This policy takes constraints from the sensor policy and generates waypoints $\mathcal{W}_t$ for shape control. The frame policy handles local obstacle avoidance and path optimization within a limited spatial horizon.

**Shape Policy ($\pi_o$):** Operates at fast timescales to control robot morphological adaptation. For deformable robots, this includes shape changes, stiffness modulation, and configuration updates. For conventional robots, this might include gait transitions, tool selection, or behavioral mode switches.

The key insight is that these policies are not manually designed hierarchies but emerge naturally from the temporal structure of the Hamiltonian dynamics. Fast variables (sensor adaptation) reach quasi-equilibrium before slower variables (shape changes) evolve significantly, creating natural scale separation without manual decomposition.

This extended analysis demonstrates that geometric structure is not merely a useful inductive bias but a necessary foundation for solving complex navigation problems that require multi-scale reasoning, online adaptation, and long-horizon stability.

## B    EXTENDED RELATED WORK SURVEY

### B.1    GEOMETRY AND MECHANICS PRIMER

Navigation learning methods can be categorized by their underlying mathematical spaces, with significant implications for performance and theoretical guarantees:

**Euclidean Space Methods ($\mathbb{R}^n$):** Standard RL treats navigation as optimization in flat spaces using Euclidean distance metrics. Enhanced PPO (Taheri et al. (2024)) demonstrate improved collision avoidance but ignore inherent geometric structure of robotic systems. Sample efficiency remains poor, typically requiring millions of environment interactions (Dehghani Tezerjani et al. (2024)).

**Lie Group Methods:** Recognition of orientation constraints has led to SE(2) and SE(3) formulations using equivariant neural architectures. These preserve rotational and translational symmetries but remain primarily limited to manipulation rather than navigation tasks.

**Riemannian Manifold Approaches:** Advanced geometric formulations employ differential geometry for constraint handling through tangent space projections. Martínez-Rubio & Pokutta (2023) demonstrates constraint satisfaction through geometric structure rather than penalty methods, achieving superior theoretical properties but limited practical deployment.

**Hamiltonian and Symplectic Methods:** Port-Hamiltonian neural networks show significant performance improvements through symplectic integrators, proving that respecting geometric structure fundamentally improves learning dynamics. However, applications remain confined to simple control problems.

### B.2    SAFETY-CRITICAL NAVIGATION TAXONOMY

**External Safety Projection:** Control Barrier Functions create safe action spaces through constraint projection. Neural Network Zeroing Barrier Functions (Feng et al. (2023)) enable collision-free navigation, while adaptive safety constraints (Mohammad & Bezzo (2025)) handle dynamic environments. Social navigation approaches (Jang & Ghaffari (2024)) extend CBFs to human-robot interaction. These methods achieve formal safety guarantees but often exhibit conservative behaviors due to the separation between safety and optimality.

**Energy-Integrated Safety:** Our approach incorporates safety directly within the Hamiltonian energy structure via barrier potentials. This enables aggressive navigation while maintaining formal guarantees through symplectic structure preservation, avoiding the conservatism of external projection methods.

### B.3    DEFORMABLE AND SOFT ROBOT NAVIGATION

**Hyperelastic Material Models:** Recent advances include pressure-stiffening control with 6.40% maximum error validation (Roshanfar et al. (2023)) and passivity-based control using differential geometry of curves (Caasenbrood et al. (2022)). Spectral Submanifold Reduction (Alora et al. (2025)) achieves computational speedup for real-time hyperelastic control with stability guarantees.

**Ring and Circular Robots:** Liquid Crystal Elastomer responses enable aerial gap navigation (Qi et al. (2024)) through predetermined actuation patterns. HAVEN (Mulvey & Nanayakkara (2024)) navigates constrained spaces via fixed shape-changing sequences based on multimodal perception. These approaches use offline parameter optimization with deterministic execution, lacking online adaptation capabilities.

**Physics-Informed Learning:** PINN-Ray (Wang et al. (2024)) achieves state-of-the-art hyperelastic displacement prediction, while extensions to non-conservative effects (Liu & Della Santina (2024)) provide experimental validation. However, these remain primarily modeling tools rather than adaptive control frameworks.

### B.4    NEURAL SCENE REPRESENTATIONS FOR NAVIGATION

**NeRF-Based SLAM:** Real-time dense reconstruction through NICE-SLAM (Zhu et al. (2022)) and keyframe-free tracking via iMAP (Sucar et al. (2021)) provide rich environmental representations.

Neural Topological SLAM (Chaplot et al. (2020)) combines learning with classical planning, while semantic approaches (Zheng et al. (2025)) integrate large vision models.

**3D Gaussian Splatting:** GS-SLAM (Yan et al. (2024)) and SplaTAM (Keetha et al. (2024)) demonstrate state-of-the-art reconstruction quality with real-time performance, offering dense 3D representations suitable for navigation applications.

**Integration with Energy Terms:** Scene representations feed our energy formulation through:

$$\text{Barrier Energy:} \quad \mathcal{U}_{\text{barrier}} = \sum_{\text{obstacles}} b(\text{SDF}(\mathbf{x})) \tag{15}$$

$$\text{Free-Space Energy:} \quad \mathcal{U}_{\text{free}} = - \sum_{\text{free regions}} w(\mathbf{x}) \tag{16}$$

$$\text{Goal Energy:} \quad \mathcal{U}_{\text{goal}} = \|\mathbf{x} - \mathbf{x}_{\text{goal}}\|^2 \tag{17}$$

### B.5 MULTI-SCALE AND HIERARCHICAL METHODS

**Hierarchical RL:** Task decomposition approaches like HRL4IN (Li et al. (2020)) handle heterogeneous navigation phases, while (Lee et al. (2023)) learns specialized policy families with high-level coordination. These require manual decomposition and struggle with principled coordination, often leading to ad-hoc design choices without theoretical guarantees.

**Multi-Agent Coordination:** RoboBallet (Lai et al. (2025)) achieves coordination for 8 robots across 40 tasks using graph neural networks. MACRPO (Kargar & Kyrki (2021)) enhances information sharing beyond parameter sharing. However, these approaches lack the geometric structure preservation critical for deformable robot coordination.

### B.6 IMITATION LEARNING FOR NAVIGATION

**Behavioral Cloning:** RT-1 (Brohan et al. (2023)) demonstrates impressive generalization across 700+ tasks using 130k demonstration episodes with transformer architectures achieving significant zero-shot performance improvements.

**Inverse Reinforcement Learning:** GAIL for Safe Navigation (Tai et al. (2018)) combines generative adversarial imitation with safety constraints. DAgger for Continuous Navigation (Patanam et al.) iteratively improves policies through expert querying.

**Sub-Optimal Demonstrations:** Confident Imitation Learning (Zhang et al. (2022)) handles demonstration uncertainty through confidence-aware training, addressing distribution shift in novel environments.

These approaches excel with high-quality demonstrations but assume expert availability and struggle with the full behavioral range needed for adaptive deformation strategies.

### B.7 FOUNDATION MODEL INTEGRATION

Large-scale models for navigation reasoning (Zhu et al. (2024); Wang et al. (2025)) focus on high-level semantic understanding and multi-agent coordination at the symbolic level. Foundation models excel at reasoning and semantic understanding, while our GRL-SNAM provides principled low-level geometric control.

**Integration Pathway:** Foundation models could generate high-level objectives encoded as potential energy terms in our energy functional $\mathcal{R}(q_t)$. The geometric structure preservation ensures high-level semantic goals translate into physically consistent behaviors, addressing the critical gap where foundation model outputs often lack grounding in physical dynamics.

### B.8 PARADIGM COMPARISON

This comprehensive survey positions GRL-SNAM as uniquely addressing the intersection of geometric structure preservation, multi-scale coordination, and deformable robot control—capabilities that existing approaches handle separately or incompletely.

Table 4: Extended paradigm-level comparison of learning frameworks. Scoring: ✓= comprehensive support, △= limited support, ×= not supported.

| Capability | GRL-SNAM | Standard RL | Geometric RL | Imitation Learning | Semi/Unsupervised | CBF Methods | Hierarchical RL | Foundation Models |
|---|---|---|---|---|---|---|---|---|
| Energy Conservation | ✓ | × | △ | × | × | × | × | × |
| Geometric Structure | ✓ | × | ✓ | × | △ | △ | × | × |
| Constraint Integration | ✓ | △ | ✓ | × | × | ✓ | △ | △ |
| Online Adaptation | ✓ | △ | △ | × | ✓ | △ | ✓ | ✓ |
| Multi-Scale Coordination | ✓ | × | × | × | × | × | ✓ | △ |
| Sample Efficiency | ✓ | × | △ | ✓ | △ | △ | △ | ✓ |
| Zero-Shot Generalization | ✓ | × | △ | × | ✓ | × | × | ✓ |
| Real-World Deployment | ✓ | ✓ | △ | ✓ | △ | ✓ | ✓ | △ |
| Deformable Robot Support | ✓ | × | × | × | × | × | × | × |

**Scoring Criteria:**

- **Energy Conservation**: Explicit conservation laws in dynamics
- **Geometric Structure**: Preservation of manifold properties
- **Constraint Integration**: Safety/task constraints within optimization
- **Online Adaptation**: Real-time policy modification during deployment
- **Multi-Scale Coordination**: Principled coordination across temporal scales
- **Sample Efficiency**: Learning with minimal environment interaction
- **Zero-Shot Generalization**: Performance in unseen environments
- **Real-World Deployment**: Practical implementation feasibility
- **Deformable Robot Support**: Explicit modeling of shape change

## B.9 KEY INSIGHTS

Our framework builds upon a set of Hamiltonian and reinforcement learning principles, unifying offline reference dynamics with online adaptive updates. Below, we summarize the six key insights that form the backbone of GRL-SNAM.

**1. Hamiltonian energy as task reward.** We define the Hamiltonian

$$\mathcal{H}(q,p) = K(p) + P(q),\tag{18}$$

with kinetic energy $K$ and task-specific potential $P$. In our setup, $P$ encodes navigation objectives (goal attraction, barrier avoidance, deformation penalties). Following Pontryagin et al. (1962); Bajaj & Nguyen (2024), the Hamiltonian coincides with the surrogate objective in policy gradient methods, i.e.

$$\nabla_\theta J(\pi_\theta) \approx \nabla_\theta \mathbb{E}_{\pi_\theta}[-\mathcal{H}(q,p)],\tag{19}$$

linking task reward to the Hamiltonian gradient flow. This equivalence grounds the DPO surrogate in a physical structure.

**2. Offline Hamiltonian vs. Online task reward.** In offline training, the agent minimizes trajectories under a fixed $\mathcal{H}$ constructed from synthetic local patches. Online, the environment is sensed, and task rewards $\mathcal{R}_{\text{env}}$ are parsed into Hamiltonian subtasks. By interpreting

$$\mathcal{H}_{\text{online}} = \mathcal{H}_{\text{offline}} + \Delta \mathcal{R}_{\text{env}},\tag{20}$$

we align local sensory updates with the reference offline Hamiltonian. This mirrors the adaptive control interpretation in Åström & Wittenmark (2010).

**3. Offline policy as reference Hamiltonian.** Every offline policy $\pi_{\text{ref}}$ is equivalent to a reference Hamiltonian $\mathcal{H}_{\text{ref}}$, where the score function $s^\theta = \nabla \mathcal{H}_{\text{ref}}$ defines canonical dynamics:

$$\dot{q} = \frac{\partial \mathcal{H}_{\text{ref}}}{\partial p}, \qquad \dot{p} = -\frac{\partial \mathcal{H}_{\text{ref}}}{\partial q}.\tag{21}$$

Online adaptation then minimizes the divergence

$$D(\pi_{\text{online}} \parallel \pi_{\text{ref}}) \propto \mathbb{E}\big[\|\nabla \mathcal{H}_{\text{online}} - \nabla \mathcal{H}_{\text{ref}}\|^2\big],\tag{22}$$

a structure exploited in score-based models (Song et al., 2021).

**4. Advantages of stagewise updates.** Rather than solving adjoint equations as in Pontryagin's Maximum Principle, we adopt a stagewise decomposition. Each stage defines a local $\mathcal{H}_t$ and is integrated feedforward:

$$\theta_{t+1} = \theta_t - \eta \nabla_\theta \mathcal{H}_t. \tag{23}$$

This avoids backward-in-time adjoint calculations and recovers the efficiency noted in adjoint-free feedforward networks (Chen et al., 2018; Kidger et al., 2021).

**5. Universality of the pipeline.** Our pipeline

$$\text{Environment} \xrightarrow{\text{Encoder}} \text{Context} \xrightarrow{\text{Setup}} \mathcal{H}_{\text{adapted}} \tag{24}$$

is universal. As long as $\mathcal{H}$ is differentiable, adaptation reduces to evaluating its gradients, regardless of whether the system is white-box (explicit potentials) or black-box (sensor-level inputs). This follows from the variational formulation of differentiable programming (Baydin et al., 2018).

**6. Navigator as meta-controller.** The navigator policy $\pi_{\text{nav}}$ interacts with three black boxes: the offline Hamiltonian $\mathcal{H}_{\text{ref}}$, the online sensed reward $\mathcal{R}_{\text{env}}$, and the adaptive fusion $\mathcal{H}_{\text{adapt}}$. Its role is to formulate and solve

$$\mathcal{H}_{\text{adapt}} = \alpha \mathcal{H}_{\text{ref}} + (1 - \alpha)\mathcal{R}_{\text{env}}, \tag{25}$$

where $\alpha$ is dynamically updated by the context encoder (e.g., LSTM). This positions the navigator as a meta-controller that continually reforms the Hamiltonian problem, a principle consistent with adaptive RL formulations in Kirk (2004).

### B.10 SECANT GAUSS–NEWTON CONTROLLER WITH IMPLICIT OBSERVABLE TRANSITION

In the online setting, policy parameters ($\Theta = [\beta, \gamma, \{\alpha_i\}]$) do not act directly on target observables $y_{\text{tgt}}$. Instead, there exists an implicit transition chain:

$$\Theta \xrightarrow{\text{policy-induced energy}} \mathcal{E}_\Theta \xrightarrow{\text{dynamics rollout}} z \mapsto f(\Theta) \xrightarrow{\text{task map}} y_{\text{tgt}}, \tag{26}$$

where $f(\Theta)$ denotes observables (e.g., clearance, progress, admissible speed) produced after symplectic updates under the reshaped Hamiltonian $\mathcal{E}_\Theta$.

**Implicit Jacobian.** The true Jacobian of observables with respect to policy parameters is

$$\frac{\partial f}{\partial \Theta} = \frac{\partial f}{\partial z} \cdot \frac{\partial z}{\partial \Theta}, \tag{27}$$

which is expensive to evaluate through full rollouts. Instead, we maintain a rank-1 secant estimate

$$\widehat{J}_t \approx \frac{f(\Theta^t) - f(\Theta^{t-1})}{\Theta^t - \Theta^{t-1}}, \tag{28}$$

smoothed via exponential moving average to reduce noise. This captures the *implicit effect* of parameter changes on observables.

**Energy Reshaping Step.** With the secant Jacobian, the Gauss–Newton update becomes

$$\Delta\Theta = -\widehat{G}^{-1}\widehat{J}^\top W\left(f(\Theta) - y_{\text{tgt}}\right), \qquad \widehat{G} = \widehat{J}^\top W \widehat{J} + \varepsilon I, \tag{29}$$

followed by projection and per-head learning rates:

$$\Theta^{t+1} = \text{Proj}_{\Theta \geq 0}\left(\Theta^t + \text{diag}(\eta_b, \eta_g, \eta_\alpha)\,\Delta\Theta\right). \tag{30}$$

Here, $W$ is a weighting matrix over observables, and $\varepsilon I$ stabilizes inversion.

**Meta-Learning View.** The Navigator maintains a meta-objective

$$\mathcal{L}_{\text{nav}}(\Theta; \phi) = \ell\left(f(\Theta; z), y_{\text{tgt}}\right), \tag{31}$$

with context-dependent parameters $\phi$ (e.g., encoder weights for $\alpha$). The implicit gradient is

$$\nabla_\Theta \mathcal{L}_{\text{nav}} = \widehat{J}^\top W\left(f(\Theta) - y_{\text{tgt}}\right), \tag{32}$$

which coincides with the secant Gauss–Newton step above. Thus, meta-learning is implemented not by direct regression on $\Theta$, but by observable alignment through the implicit policy $\rightarrow$ observables $\rightarrow$ targets chain.

**Interpretation.** This formulation clarifies that the controller does not operate in parameter space alone. Instead, it continuously reshapes Hamiltonian parameters so that the *induced observables* approach task targets, effectively coupling multiple energy components (safety, progress, speed) without additional rollouts.

**Sequential Query–Response (as ports).** At each $t$ the Navigator issues queries $\mathcal{Q}_k^t$ and receives responses $\mathcal{R}_k^t$, which determine $y_{\text{tgt}}(\mathcal{R}^t)$ and any weights in $\mathbf{W}$; the update equation **??** (with the secant $\widehat{\mathbf{J}}$) is then applied to each block $k \in \{y, f, o\}$:

$$\theta_k^{t+1} = \theta_k^t + h\,\Delta\theta_k^t, \quad \Delta\theta_k^t = -\,\widehat{\mathbf{G}}_k^{-1}\widehat{\mathbf{J}}_k^{\top}\mathbf{W}_k\big(f_k(\theta_k^t) - y_{\text{tgt},k}(\mathcal{R}_k^t)\big).$$

**State Evolution.** With updated parameters, each policy advances its state as before,

$$z_k^{t+1} = z_k^t + \tau_k\,J_k\,s_k^{\theta_k^{t+1}}(z_k^t, \mathcal{C}_t^{\text{updated}}, t).$$

*Remark.* If a strict port-Hamiltonian view is desired, the $\mathbf{J}^{\top}\mathbf{W}(y_{\text{tgt}} - f(\Theta))$ term enters $\dot{\Pi}$ as an external port (input) rather than being baked into the potential; the resulting discrete update is identical to the secant Gauss–Newton step above.

## C  HYPERELASTIC RING ROBOT MODEL

We model the deformable robot as a closed hyperelastic ring with reduced-order dynamics to enable efficient navigation while capturing essential deformation behaviors.

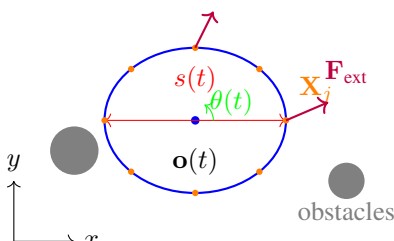

Figure 6: Hyperelastic ring robot model showing generalized coordinates $(s, \mathbf{o}, \theta)$, spline sample points $\mathbf{X}_j$, and external force fields.

### C.1  GEOMETRIC REPRESENTATION

The robot boundary is defined by a periodic cubic B-spline curve with $n_{\text{ctrl}}$ control points:

$$\mathbf{S}(u) = \sum_{i=1}^{n_{\text{ctrl}}} N_{i,3}(u)\mathbf{P}_i, \quad u \in [0, 1] \tag{33}$$

where $N_{i,3}(u)$ are degree-3 B-spline basis functions with $C^2$ continuity. The base shape is a unit circle:

$$\mathbf{P}_{0,i} = r_{\text{base}} \begin{bmatrix} \cos(2\pi i/n_{\text{ctrl}}) \\ \sin(2\pi i/n_{\text{ctrl}}) \end{bmatrix} \tag{34}$$

World coordinates are computed via similarity transformation:

$$\mathbf{P}_i(t) = \mathbf{o}(t) + s(t)\mathbf{R}(\theta(t))\mathbf{P}_{0,i} \tag{35}$$

where $s(t)$ is uniform scale, $\mathbf{o}(t) \in \mathbb{R}^2$ is center position, $\theta(t)$ is orientation.

For physics computation, we sample $K$ points on the curve using B-spline evaluation matrix $B \in \mathbb{R}^{K \times n_{\text{ctrl}}}$:

$$\mathbf{X}_j = \sum_{i=1}^{n_{\text{ctrl}}} B_{ji}\mathbf{P}_i, \quad j = 1, \ldots, K \tag{36}$$

---

**Algorithm 1** Hyperelastic Ring Deformation Policy

---

1: **Input:** State $(s, \dot{s}, \mathbf{o}, \dot{\mathbf{o}}, \theta, \omega)$, obstacles $\{(\mathbf{c}_k, r_k)\}$, target $\mathbf{x}_{\text{target}}$
2: **Output:** Updated state $(s', \dot{s}', \mathbf{o}', \dot{\mathbf{o}}', \theta', \omega')$
3: Update geometry: $\mathbf{X}_j \leftarrow$ sample curve at current state
4: Compute distances: $d_{jk} \leftarrow \|\mathbf{X}_j - \mathbf{c}_k\| - r_k$, clearance: $d_{\min} \leftarrow \min_{j,k} d_{jk}$
5:                                                         ▷ — Conservative Forces —
6: IPC barriers: $\mathbf{g}_j \leftarrow \sum_k \frac{\partial b_{\text{IPC}}(d_{jk})}{\partial \mathbf{X}_j}$
7: Adaptive bulk: $F_{s,\text{bulk}} \leftarrow -\frac{\partial \mathcal{U}_{\text{bulk}}}{\partial s}$ with $A_{\text{target}}(d_{\min})$
8:                                              ▷ — Non-Conservative Forces —
9: Stage forces: $\mathbf{F}_{\text{stage},j} \leftarrow$ goal + radial + tangential components
10: Friction: $\mathbf{F}_{\text{friction},j} \leftarrow -\mu \text{contact\_pressure} \cdot \text{tangent\_velocity}$
11:                                         ▷ — Generalized Force Mapping —
12: Map to coordinates: $F_s, \mathbf{F}_o, \tau \leftarrow$ virtual work from $\{\mathbf{g}_j + \mathbf{F}_{\text{stage},j} + \mathbf{F}_{\text{friction},j}\}$
13:                                                   ▷ — Integration —
14: Update velocities: $\dot{s}' \leftarrow \dot{s} + \Delta t \cdot F_s / M_s$, etc.
15: Update positions: $s' \leftarrow \text{clamp}(s + \Delta t \cdot \dot{s}')$, $\mathbf{o}' \leftarrow \mathbf{o} + \Delta t \cdot \dot{\mathbf{o}}'$, etc.
16: **return** updated state

---

## C.2 ENERGY FORMULATION

The total Hamiltonian combines kinetic and potential components:

$$\mathcal{H} = \frac{1}{2} M_s \dot{s}^2 + \frac{1}{2} M_o \|\dot{\mathbf{o}}\|^2 + \frac{1}{2} I \omega^2 + \mathcal{U}_{\text{barrier}} + \mathcal{U}_{\text{bulk}} \tag{37}$$

**IPC Barrier Energy:** Collision avoidance using Incremental Potential Contact barriers:

$$\mathcal{U}_{\text{barrier}} = \sum_{j=1}^{K} w_j \ell_j \sum_{k=1}^{N_{\text{obs}}} b_{\text{IPC}}(d_{jk}) \tag{38}$$

where $w_j = 1/K$, $\ell_j = \|\mathbf{X}_j'\|$, $d_{jk}$ is distance from sample $j$ to obstacle $k$:

$$b_{\text{IPC}}(d) = \begin{cases} -(d - \hat{d})^2 (\log d - \log \hat{d}) & \text{if } 0 < d < \hat{d} \\ 0 & \text{if } d \geq \hat{d} \\ V_{\text{penalty}} & \text{if } d \leq 0 \end{cases} \tag{39}$$

**Adaptive Bulk Energy:** Area conservation with clearance-dependent target:

$$\mathcal{U}_{\text{bulk}} = \frac{k_{\text{bulk}}}{2} (A(s) - A_{\text{target}})^2 \tag{40}$$

where $A(s) = s^2 A_{\text{ref}}$ and:

$$A_{\text{target}} = [\alpha + (1 - \alpha) \tanh(\beta \cdot \max(d_{\min}, 0))] A_{\text{ref}} \tag{41}$$

with $\alpha = 0.25$, $\beta = 2.5$ encouraging compression in tight spaces.

## C.3 GENERALIZED FORCE MAPPING

Forces on spline samples map to generalized coordinates via virtual work:

$$F_s = -\frac{\partial \mathcal{U}}{\partial s} + \sum_{j=1}^{K} \mathbf{F}_{\text{ext}}(\mathbf{X}_j) \cdot \frac{\partial \mathbf{X}_j}{\partial s} - \gamma_s \dot{s} \tag{42}$$

$$\mathbf{F}_o = -\frac{\partial \mathcal{U}}{\partial \mathbf{o}} + \sum_{j=1}^{K} w_j \mathbf{F}_{\text{ext}}(\mathbf{X}_j) - \gamma_o \dot{\mathbf{o}} \tag{43}$$

$$\tau = -\frac{\partial \mathcal{U}}{\partial \theta} + \sum_{j=1}^{K} w_j \mathbf{F}_{\text{ext}}(\mathbf{X}_j) \cdot (\mathbf{J}(\mathbf{X}_j - \mathbf{o})) - \gamma_\theta \omega \tag{44}$$

where $\mathbf{J} = \begin{bmatrix} 0 & -1 \\ 1 & 0 \end{bmatrix}$ generates rotation and $\frac{\partial \mathbf{X}_j}{\partial s} = \mathbf{R}(\theta) \mathbf{P}_{0,j}$.

## C.4 Navigator as Meta-Hamiltonian Learner

The GRL-SNAM operates as a meta-learning system that coordinates multi-scale policies by learning how to update their Hamiltonian energy functions rather than directly manipulating phase space states.

**Navigation specialization.** Throughout the paper we further simplify our navigator's Hamiltonian surrogate by assuming fixed $\omega$ throughout all environment when we primarily focus on learning to optimize the environment-indexed linear cone generated by task energies. The potential energy is denoted as follows and is shown in Figure 5:

$$\mathcal{R}(q; \eta(\mathcal{E})) = \underbrace{\|\boldsymbol{y}\|_S^2}_{\text{Sensor Cost } (E_{sensor})} + \underbrace{\beta\|\boldsymbol{c} - \mathbf{x}_g\|_2^2}_{\text{Goal Attraction } (E_{goal})}$$
$$+ \underbrace{\lambda E_{obj}(q(t))}_{\text{Deformation Energy } (E_{obj})} + \underbrace{\sum_{i \in \mathcal{C}_t(\mathcal{E}, q)} \alpha_i b(d_i, \hat{d})}_{\text{Collision Barriers (FPE)}}. \tag{45}$$

The *Sensor Policy* contributes the sensor cost $E_{sensor}$, regularizing information acquisition. The *FPE (Free Path Extractor)* governs goal attraction and path-planning in current environment $\mathcal{E}$ via goal attraction force $E_{goal}$ and barrier potentials, balancing reachability with safety. The *Reconfig Policy* governs deformation, enabling radius modulation for narrow passages. This decomposition highlights that the total Hamiltonian is not a monolithic reward but a structured sum of physically interpretable energies, each attached to a specialized policy. In addition, we assume friction and additional forces (e.g. derived from safety constraints) are happened in FPE submodule only, and we pick a particular parametrization as:

$$\begin{aligned} \Gamma_y^\xi &\equiv 0, \quad G_y^\xi \equiv 0, \quad u_y^\xi \equiv 0, \\ \Gamma_f^\xi &= \mu_\xi(\mathcal{E})\mathbf{I}, \quad G_f^\xi = \mathbf{I}, \quad u_f^\xi \neq 0, \\ \Gamma_o^\xi &\equiv 0, \quad G_o^\xi \equiv 0, \quad u_o^\xi \equiv 0, \end{aligned} \tag{46}$$

Thus the meta policy $g_\xi$ *directly* produces the cone coordinates of potentials and non-conservative forces correction in FPE module as a friction and a port correction term:

$$g_\xi(\mathcal{E}) = \left[\, \eta_\xi(\mathcal{E}), \mu_\xi(\mathcal{E}), u_f^\xi \,\right],$$

which defines the potential—and therefore the generalized force —for the stagewise motion-planning Hamiltonian rollouts. Learning $g_\xi$ gives a *meta-policy* that maps environments to energy weights, i.e. a stagewise bilevel scheme where the inner layer optimizes motion under $H_k$ and the outer layer trains $\xi$ so that scenario-level QoIs/constraints are satisfied across environments.

**Learning $\eta_\xi(\mathcal{E}), \mu_\xi(\mathcal{E})$ throughout "Module→surrogate reduction".** Each submodule exposes a response map that provides integrated dynamics rollout and additional quantity of interests (QoI) as the feedback:

$$\mathsf{R}_k : \left(H_k; \theta_k, \mathcal{E}, \xi\right) \longmapsto \{z_t^{(k)}, s_t^{(k)}, \mathrm{QoI}_k\}_{t=0}^{T_k}, \qquad k \in \mathcal{K},$$

where $T_k$ refers time scale for each submodule. To be more specific:

$$\mathrm{QoI}_y = \Delta\mathcal{E}, \quad \mathrm{QoI}_f = \{v_t^{(f)} := M_f^{-1} p_t^{(f)}\}_{t=0}^{T_f}, \quad \mathrm{QoI}_o = \{\min_{i \in \mathcal{C}_t(\mathcal{E}, q)} d_i(q_t^{(o)}; \mathcal{E})\}_{t=0}^{T_o}. \tag{47}$$

Namely, feedback QoIs are: environmental update, velocity observation, and min distance clearance (and thus collision violation). Additional QoIs, that can be deduced from $z_t^{(k)}, s_t^{(k)}$, are not stated here explicitly. To train a policy that output $\eta(\mathcal{E})$ and $\mu_\xi(\mathcal{E})$ with different environments, we propose to minimize

$$\mathcal{L}(\eta_\xi, \mu_\xi) = \mathbb{E}_\mathcal{E}\left[w_q\|\boldsymbol{q} - \boldsymbol{q}_{ref}\|_2^2 + w_v\|\boldsymbol{v} - \boldsymbol{v}_{ref}\|_2^2 + w_\mu\|\mu - \mu_{ref}\|_2^2 + w_d\,\mathcal{L}_{\text{multi}}\right]. \tag{48}$$

---

**Algorithm 2** Meta-policy training of $\eta(\mathcal{E})$ and $\mu_\xi(\mathcal{E})$

---

**Require:** dataset of different environments $\{\mathcal{E}, \alpha_{ref}, \beta_{ref}, \lambda_{ref}, \mu_{ref}\} \in \mathcal{D}$, step range $T \in \{2, 3, 4, 5, 6\}$, short rollout trials $M$, time step $\Delta t$, weights $w_\bullet$
 1: Initialize $\xi$
 2: **for** epoch $= 1, 2, \dots$ **do**
 3:      **for** batch $\mathcal{B} \subset \mathcal{D}$ **do**
 4:          Sample $(q_0, p_0)$ per scene $\mathcal{E}$ in $\mathcal{B}$
 5:          Build **tokens** from active constraints $\mathcal{C}_t(\mathcal{E}, q_0)$, goal $\boldsymbol{x}_g$, and current $(q_0, p_0)$
 6:          $(\{\alpha_j\}, \beta) \leftarrow \eta_\xi(\textbf{tokens}), \quad \mu \leftarrow \mu_\xi(\textbf{tokens})$
 7:          **for** $m = 1, 2, \dots, M$ **do**
 8:              Sample a near obstacle point $\tilde{q}_0$, initialize $\tilde{p}_0$ towards nearest obstacle at $\tilde{q}_0$.
 9:              Integrate equation 8 and equation 9 for $T$ steps from $(\tilde{q}_0, \tilde{p}_0)$.
10:                                        $\triangleright$ e.g. Symplectic Euler
11:              Compute $clr \leftarrow \min_{i,t} d_i(\tilde{q}_t; \mathcal{E})$
12:              Compute $\mathcal{L}_{\text{multi}} \leftarrow \mathcal{L}_{multi} + \frac{1}{M} \text{softplus}\left(\frac{r_{min} - \text{clr}}{\hat{d}}\right)$.
13:                            $\triangleright$ $r_{min}$ refers minimal radius of robot object.
14:          **end for**
15:          Integrate equation 8 and equation 9 for $T_k$ steps from $(q_0, p_0)$
16:          Evaluate loss $\mathcal{L}(\eta_\xi, \mu_\xi)$ in equation 48
17:          Update $\eta_\xi, \mu_\xi$ using $\nabla_{\eta_\xi}\mathcal{L}$ and $\nabla_{\mu_\xi}\mathcal{L}$
18:          E.g. Adam Optimizer
19:      **end for**
20: **end for**

---

where $\mathcal{L}_{\text{multi}}$ is a short multi-start robustness penalty that re-rolls from perturbed $q_t^{(k)}$ seeds near obstacles to discourage brittle $\eta_\xi(\mathcal{E})$ and $\mu_\xi(\mathcal{E})$ (details are addressed in Algorithm 2), and $w_\bullet \geq 0$ are user-input hyperparameters. The training via equation 48 can be conducted offline, component-wise, or even fine-tuned online, but we state that it is important to fully utilize the instantaneous response from a real navigation scenario which provides the scheme of per-scneario online correction even when $\eta_\xi(\mathcal{E})$ and $\mu_\xi(\mathcal{E})$ are properly trained under large-scale simulated dataset with reference potentials.

**Online Adaptation of** $g_\xi(\mathcal{E})$ **via QoIs**    We state how response map for each environment can yield a correction term under online navigation scenario. Given response $\mathsf{R}_k$ at time $t$ we construct an *observable* measurement vector and its reference goal:

$$
y_t = \begin{bmatrix} -\text{clr}_t \\ \text{dist}_t \\ -\text{speed}_t \end{bmatrix} \in \mathbb{R}^3, \qquad y_t^\star = \begin{bmatrix} -m_{\text{safe}} \\ \text{dist}_t - \varepsilon_{\text{prog}} \\ -\max\left(\text{speed}_t, \mathbf{1}_{\{\text{clr}_t \geq m_{\text{safe}}\}} v_{\text{min}}\right) \end{bmatrix},
$$

where clr is the minimum clearance to inflated obstacles, dist is goal distance (to the global goal point), and speed $= \|v\|$. We update only the *active* barrier weights by selecting an index set $\mathcal{I}_t$ that represents nearby obstacles. Define the parameter vector

$$
\tilde{\eta}_t = \begin{bmatrix} \beta_t \\ \lambda_t \\ \alpha_{t,\mathcal{I}_t} \end{bmatrix} \in \mathbb{R}_+^{2+|\mathcal{I}_t|}, \quad \zeta_t = [\tilde{\eta}_t, \mu_t].
$$

We denote an estimator of Jabocian $J_t = \frac{\partial y}{\partial \zeta_t}$ as

$$
\widetilde{J}_t = \frac{(y_t - y_{t-1})(\zeta_t - \zeta_{t-1})^\top}{\|\zeta_t - \zeta_{t-1}\|_2^2 + \varepsilon}, \qquad J_t = \rho J_{t-1} + (1 - \rho)\widetilde{J}_t,
$$

with smoothing $\rho \in [0, 1)$ and $\varepsilon > 0$. Then, given the desired observable change $\Delta y_t^{\text{des}} := y_t^\star - y_t$, one can upate meta-policy parameter via a Tiknohov-regularized least square steps:

$$
\Delta \zeta_t = \arg\min_{\Delta\zeta} \|J_t \Delta\zeta - \Delta y_t^{\text{des}}\|_2^2 + \lambda \|\Delta\zeta\|_2^2, \quad \Rightarrow \quad \Delta\zeta_t = (J_t^\top J_t + \lambda I)^{-1} J_t^\top \Delta y_t^{\text{des}}.
$$

This yields updated $\beta_{t+1}$, $\gamma_{t+1}$, and $\alpha_{t+1,i}$ for $i \in \mathcal{I}_t$ (inactive weights keep their previous values).

$$\zeta_{t+1} = \Pi_{\mathbb{R}_+}\left((1 - \boldsymbol{\kappa})\zeta_t + \boldsymbol{\kappa}\,\Delta\zeta_t\right), \quad \boldsymbol{\kappa}_i \in [0, 1).$$

Since the update is a result of least square step, there exists residual:

$$r_t = \Delta y_t^{\mathrm{des}} - J_t\,\Delta\zeta_t.$$

The online port correction term can amend the energy change by solving another least square problem given port–observable sensitivity $P_t \approx \partial y / \partial u_f$:

$$u_{f,t}^\xi = \arg\min_{u \in \mathcal{U}} \left\|P_t u - r_t\right\|_2^2 + \lambda_u \|u\|_2^2 = (P_t^\top P_t + \lambda_u I)^{-1} P_t^\top r_t,$$

followed by componentwise clipping to a feasible box $\mathcal{U}$. A simple choice is to use $P_t = \mathrm{diag}(0, 0, \kappa_v)$ so that the port primarily regulates speed while the energy weights steer clearance and goal progress; richer $P_t$ can be learned online by the same secant recipe as $J_t$.

## D  DOMAIN-SPECIFIC POLICY IMPLEMENTATIONS

### D.1  SENSOR POLICY ($\pi_y$) DETAILS

The sensor policy maintains spatial index $\mathcal{T}_y$ of observations $(\mathbf{x}_i, \mathrm{type}_i, \mathrm{attr}_i)$ and derives three energy components from single neighbor queries:

**Barrier Potential:** Repulsion from obstacles

$$b_\Sigma(z_y, \mathcal{C}_t) = \sum_{i \in \mathcal{N}_{\mathrm{obs}}} w_i \exp\left(-\frac{\|\mathbf{x}_i - \boldsymbol{c}_y\|^2}{2\sigma_b^2}\right) \tag{49}$$

**Free-Space Potential:** Attraction to open regions

$$\mathcal{V}_{\mathrm{free}}(z_y, \mathcal{C}_t) = -\sum_{j \in \mathcal{N}_{\mathrm{free}}} w_j \exp\left(-\frac{\|\mathbf{x}_j - \boldsymbol{c}_y\|^2}{2\sigma_f^2}\right) \tag{50}$$

**Density Potential:** Information-theoretic density measure

$$\rho(z_y, \mathcal{C}_t) = -\sum_{k \in \mathcal{N}_{\mathrm{all}}} w_k \log\left(1 + \frac{n_k}{|\mathcal{N}_{\mathrm{all}}|}\right) \tag{51}$$

The complete sensor score function is:

$$s_y^{\theta_y}(z_y, \mathcal{C}_t, t) = \nabla_{z_y} \left[\frac{1}{2}\|p_y\|_{M_y^{-1}}^2 + \alpha_b b_\Sigma + \alpha_f \mathcal{V}_{\mathrm{free}} + \alpha_d \rho\right] \tag{52}$$

### D.2  FRAME POLICY ($\pi_f$) DETAILS

The frame policy uses $\mathcal{T}_f$ storing path samples with safety/contact distances and goal influence:

**Safety Field:** Distance-based safety measure

$$S(z_f, \mathcal{C}_t) = \sum_{i=1}^{N_s} w_i \max(0, d_{\mathrm{safe}}^{\mathrm{threshold}} - d_{\mathrm{safe}}^i)^2 \tag{53}$$

**Contact Field:** Proximity to obstacles

$$C(z_f, \mathcal{C}_t) = \sum_{i=1}^{N_c} w_i \exp\left(-\frac{(d_{\mathrm{contact}}^i)^2}{2\sigma_c^2}\right) \tag{54}$$

**Goal Field:** Directional bias toward target

$$G(z_f, \mathcal{C}_t) = -\|\boldsymbol{c}_f - \mathbf{x}_g\|^2 + \sum_{i=1}^{N_g} w_i g_i \cos(\theta_i) \tag{55}$$

The frame score function integrates these fields:

$$s_f^{\theta_f}(z_f, \mathcal{C}_t, t) = \nabla_{z_f} \left[ \frac{1}{2} \|p_f\|_{M_f^{-1}}^2 + \alpha_s S + \alpha_c C + \alpha_g G \right] \tag{56}$$

## D.3 SHAPE POLICY ($\pi_o$) DETAILS

The shape policy controls deformation through reduced coordinates $z_o = (s, \dot{s}, \mathbf{o}, \dot{\mathbf{o}}, \theta, \omega)$:

**Smoothness Energy:** Curvature regularization

$$\mathcal{E}_{\text{smooth}} = \int_0^1 \|\kappa(u)\|^2 du \approx \sum_{j=1}^{K} w_j \|\kappa_j\|^2 \tag{57}$$

**Stretching Energy:** Arc length preservation

$$\mathcal{E}_{\text{stretch}} = \int_0^1 (\|\mathbf{S}'(u)\| - \ell_{\text{ref}})^2 du \approx \sum_{j=1}^{K} w_j (\ell_j - \ell_{\text{ref}})^2 \tag{58}$$

**Target Energy:** Configuration constraints

$$\mathcal{E}_{\text{target}} = \|\mathbf{P} - \mathbf{P}_{\text{target}}\|_F^2 + \|(s, \mathbf{o}, \theta) - (s_{\text{target}}, \mathbf{o}_{\text{target}}, \theta_{\text{target}})\|^2 \tag{59}$$

The complete shape score function is:

$$s_o^{\theta_o}(z_o, \mathcal{C}_t, t) = \nabla_{z_o} \left[ \frac{1}{2} M_s \dot{s}^2 + \frac{1}{2} M_o \|\dot{\mathbf{o}}\|^2 + \frac{1}{2} I \omega^2 + \alpha_{sm} \mathcal{E}_{\text{smooth}} + \alpha_{st} \mathcal{E}_{\text{stretch}} + \alpha_{tg} \mathcal{E}_{\text{target}} \right] \tag{60}$$

## E ALGORITHM IMPLEMENTATION DETAILS

This section provides comprehensive implementation details for Algorithm 3 presented in §3.1. We explain each algorithmic component, including parameter initialization, policy query protocols, Hamiltonian composition, integration schemes, observable extraction procedures, and online adaptation mechanisms. These details are essential for reproducible implementation but are deferred to the appendix to maintain focus on the core methodological contributions in the main text.

### E.1 INITIALIZATION

**Robot State.** Initialize phase-space coordinates $z_0 = (q_0, p_0)$ where $q_0 = (\boldsymbol{c}_0, \theta_0, s_0, \psi_0)$ contains the starting position $\boldsymbol{c}_0$, orientation $\theta_0$, scale $s_0 = 1.0$, and shape parameters $\psi_0$ (e.g., B-spline control points initialized as a circle). The initial momentum is $p_0 = 0$.

**Time Step and Horizons.** The integration time step $\tau$ is typically set to $0.01$–$0.05$ seconds. Query horizons $(T_y, T_f, T_o)$ determine how frequently each policy is queried: sensor policy every $T_y$ steps, frame policy every $T_f$ steps, and shape policy every step (or every $T_o$ steps). Typical values: $T_y = 10, T_f = 5, T_o = 1$.

**Maximum Time Horizon.** $T_{\max}$ is the maximum number of integration steps allowed before timeout, typically $5000$–$10000$ steps corresponding to $50$–$500$ seconds of simulated time.

---

**Algorithm 3** Online GRL–SNAM: Navigator-Driven Hamiltonian Composition

1: **Input:** Goal $\mathbf{x}_g$, initial $z_0 = (q_0, 0)$, step $\tau$, horizons $(T_y, T_f, T_o)$        ▷ §E.1
2: **Init:** $t \leftarrow 0$, $\mathcal{C}_0 \leftarrow \emptyset$, meta-policy $g_\xi$, Jacobians $J_0, P_0$        ▷ §E.2
3: **while** $\neg\text{REACHEDGOAL}(\boldsymbol{c}_t, \mathbf{x}_g)$ **and** $t < T_{\max}$ **do**        ▷ §E.3
4:      **Query policies & collect responses:**        ▷ §E.4
5:      **if** $t \equiv 0 \pmod{T_y}$ **then**
6:          $\mathsf{R}_y^t \leftarrow \pi_y(H_y; \theta_y, \mathcal{E}, \xi)$ returns $\{z_{0:T_y}^{(y)}, s_{0:T_y}^{(y)}, \Delta\mathcal{E}\}$        ▷ §E.5
7:          $\mathcal{E} \leftarrow \mathcal{E} \cup \Delta\mathcal{E}; \mathcal{C}_t \leftarrow \{i \mid d_i(q_t; \mathcal{E}) \leq \hat{d}\}$
8:      **end if**
9:      **if** $t \equiv 0 \pmod{T_f}$ **then**
10:          $\mathsf{R}_f^t \leftarrow \pi_f(H_f; \theta_f, \mathcal{E}, \xi)$ returns $\{z_{0:T_f}^{(f)}, s_{0:T_f}^{(f)}, \{v_\ell^{(f)}\}_{\ell=0}^{T_f}\}$        ▷ §E.6
11:      **end if**
12:      $\mathsf{R}_o^t \leftarrow \pi_o(H_o; \theta_o, \mathcal{E}, \xi)$ returns $\{z_{0:T_o}^{(o)}, s_{0:T_o}^{(o)}, \{\min_i d_i(q_\ell^{(o)}; \mathcal{E})\}_{\ell=0}^{T_o}\}$        ▷ §E.7

13:      **Meta-policy proposal:**        ▷ §E.8
14:      Build tokens $\mathcal{T}_t$ from $(\mathcal{C}_t, q_t, \mathbf{x}_g)$
15:      $[\eta_\xi^t, \mu_\xi^t, u_f^{\xi,t}] \leftarrow g_\xi(\mathcal{T}_t; \xi)$ with $\eta_\xi^t = (\beta^t, \lambda^t, \{\alpha_i^t\}_{i \in \mathcal{C}_t})$

16:      **Compose surrogate Hamiltonian:**        ▷ §E.9
         $\mathcal{R}(q_t; \eta_\xi^t, \mathcal{E}) = E_{\text{sensor}} + \beta^t E_{\text{goal}} + \lambda^t E_{\text{obj}} + \sum_{i \in \mathcal{C}_t} \alpha_i^t b(d_i)$
         $H(q_t, p_t; \omega, \xi, \mathcal{E}) = \frac{1}{2} p_t^\top M(q_t)^{-1} p_t + \mathcal{R}(q_t; \eta_\xi^t, \mathcal{E})$

17:      **Integrate dynamics:**        ▷ §E.10
18:      $\nabla_p H|_t = M(q_t)^{-1} p_t$; $\nabla_q H|_t = \nabla_q E_{\text{sensor}} + \beta^t \nabla_q E_{\text{goal}} + \lambda^t \nabla_q E_{\text{obj}} + \sum_{i \in \mathcal{C}_t} \alpha_i^t \nabla_q b(d_i)$
19:      $p_{t+\tau} = p_t - \tau \nabla_q H|_t - \tau \mu_\xi^t \nabla_p H|_t + \tau u_f^{\xi,t}$
20:      $q_{t+\tau} = q_t + \tau M(q_t)^{-1} p_{t+\tau}$

21:      **Extract observables:**        ▷ §E.11
22:      $\text{clr}_t = \min\{\min_i d_i(q_{t+\tau}; \mathcal{E}), \min_{\ell,i} d_i(q_\ell^{(o)}; \mathcal{E})\}$; $\text{dist}_t = \|\boldsymbol{c}_{t+\tau} - \mathbf{x}_g\|$; $\text{speed}_t = \|M^{-1} p_{t+\tau}\|$
23:      $y_t = [-\text{clr}_t, \text{dist}_t, -\text{speed}_t]^\top$;    $y_t^\star = [-m_{\text{safe}}, \text{dist}_t - \varepsilon_{\text{prog}}, -\max(\text{speed}_t, \mathbf{1}_{\{\text{clr}_t \geq m_{\text{safe}}\}} v_{\min})]^\top$

24:      **Parameter adaptation:**        ▷ §E.12
25:      Select $\mathcal{I}_t \subset \mathcal{C}_t$; $\zeta_t = [\beta^t, \lambda^t, \{\alpha_i^t\}_{i \in \mathcal{I}_t}, \mu_\xi^t]^\top$
26:      $\widetilde{J}_t = (y_t - y_{t-\tau})(\zeta_t - \zeta_{t-\tau})^\top / (\|\zeta_t - \zeta_{t-\tau}\|^2 + \varepsilon)$; $J_t = \rho J_{t-\tau} + (1-\rho)\widetilde{J}_t$
27:      $\Delta y_t^{\text{des}} = y_t^\star - y_t$; $\Delta\zeta_t = (J_t^\top J_t + \lambda_\zeta I)^{-1} J_t^\top \Delta y_t^{\text{des}}$
28:      $\zeta_{t+\tau} = \Pi_{\mathbb{R}_+}((1 - \boldsymbol{\kappa}) \odot \zeta_t + \boldsymbol{\kappa} \odot (\zeta_t + \Delta\zeta_t))$
29:      Unpack to $[\beta^{t+\tau}, \lambda^{t+\tau}, \{\alpha_i^{t+\tau}\}_{i \in \mathcal{I}_t}, \mu_\xi^{t+\tau}]$

30:      **Port correction:**        ▷ §E.13
31:      $r_t = \Delta y_t^{\text{des}} - J_t \Delta\zeta_t$; $u_f^{\xi,t+\tau} = \text{clip}((P_t^\top P_t + \lambda_u I)^{-1} P_t^\top r_t, \mathcal{U})$
32:      $\widetilde{P}_t = (y_t - y_{t-\tau})(u_f^{\xi,t} - u_f^{\xi,t-\tau})^\top / (\|u_f^{\xi,t} - u_f^{\xi,t-\tau}\|^2 + \varepsilon)$; $P_t = \rho_P P_{t-\tau} + (1 - \rho_P)\widetilde{P}_t$
33:      $t \leftarrow t + \tau$
34: **end while**
35: **Return:** Trajectory $\{z_\ell\}_{\ell=0}^T$, parameters $\{\eta_\xi^\ell, \mu_\xi^\ell, u_f^{\xi,\ell}\}_{\ell=0}^T$        ▷ §E.14

---

## E.2 PARAMETER STRUCTURE AND INITIALIZATION

**Meta-Policy Network.** The meta-policy $g_\xi : \mathcal{E} \to [\eta_\xi, \mu_\xi, u_f^\xi]$ can be either:

- A learned neural network trained offline via Algorithm 2
- Fixed constant values: $\beta^0 = 2.0, \lambda^0 = 1.0, \alpha_i^0 = 1.5, \mu^0 = 0.1$

If using a learned network, $\xi$ are the network parameters. If using fixed values, $g_\xi$ simply returns constants.

**Jacobian Initialization.** Initialize $J_0 \in \mathbb{R}^{3 \times (2+|\mathcal{C}_0|+1)}$ as a small random matrix or identity-scaled matrix. Initialize port Jacobian $P_0 \in \mathbb{R}^{3 \times \dim(u_f)}$ similarly. These will be refined online via secant updates.

**Environment and Context.** Initial environment $\mathcal{E}_0$ contains known static obstacles. Active constraint set $\mathcal{C}_0 = \emptyset$ starts empty and will be populated by the sensor policy's first query.

## E.3 TERMINATION CONDITIONS

The algorithm terminates when any of the following conditions is met:

**Success: Goal Reached.** $\|c_t - \mathbf{x}_g\| < \epsilon_{\text{goal}}$ where $\epsilon_{\text{goal}} = 0.05$ m (5 cm tolerance).

**Failure: Timeout.** $t \geq T_{\max}$ without reaching the goal.

**Failure: Collision.** $\min_i d_i(q_t; \mathcal{E}) < 0$, indicating penetration into an obstacle.

**Failure: Stuck.** $\|c_t - c_{t-50\tau}\| < \epsilon_{\text{stuck}}$ where $\epsilon_{\text{stuck}} = 0.01$ m, indicating the robot has not moved more than 1 cm in the last 50 steps (indicating entrapment in a local minimum).

## E.4 HIERARCHICAL QUERY FLOW

Policies are queried in a specific sequence with information flowing forward:

**Sensor Policy Query (Line 5).** Executed every $T_y$ steps. Takes as input the previous context $\mathcal{C}_{t-T_y}$, current state $z_y^t = (c_t, \dot{c}_t)$, and goal $\mathbf{x}_g$. Performs ray-casting from the robot's position to detect new obstacles within sensing range.

**Frame Policy Query (Line 8).** Executed every $T_f$ steps. Takes as input the updated context $\mathcal{C}_t$ from the sensor policy, current full state $z_f^t = (q_t, p_t)$, and goal $\mathbf{x}_g$. Plans a short-horizon path considering known obstacles.

**Shape Policy Query (Line 10).** Executed every step. Takes as input the context $\mathcal{C}_t$ from sensor and the current shape state $z_o^t = (s_t, \psi_t, \dot{s}_t, \dot{\psi}_t)$. Computes deformation forces to navigate through constrained spaces.

**Information Flow.** $\boxed{\text{Sensor}} \xrightarrow{\mathcal{C}_t} \boxed{\text{Frame}} \xrightarrow{\text{implicit}} \boxed{\text{Shape}} \xrightarrow{\text{all responses}} \boxed{\text{Navigator}}$

## E.5 SENSOR RESPONSE

**Response Structure.** $\mathsf{R}_y^t = \{z_{0:T_y}^{(y)}, s_{0:T_y}^{(y)}, \Delta\mathcal{E}\}$ where:

- $z_{0:T_y}^{(y)} = \{(q_\ell^{(y)}, p_\ell^{(y)})\}_{\ell=0}^{T_y}$ is the integrated trajectory of the sensor policy's local Hamiltonian $H_y$
- $s_{0:T_y}^{(y)} = \{s_\ell^{(y)}\}_{\ell=0}^{T_y}$ is the score function (dynamics drift) at each step

- $\Delta\mathcal{E}$ contains newly detected obstacles: $\Delta\mathcal{E} = \{(\mathbf{x}_j, r_j, \text{type}_j)\}$ with positions, radii, and types

**Environment Update (Line 6).** Merge new detections: $\mathcal{E} \leftarrow \mathcal{E} \cup \Delta\mathcal{E}$. Update active constraint set: $\mathcal{C}_t \leftarrow \{i \mid d_i(q_t; \mathcal{E}) \leq \hat{d}\}$ where $\hat{d}$ is the sensing/activation radius (typically 0.5–1.0 m).

**Policy Hamiltonian (not directly observed).** The sensor policy operates on $H_y(q_y, p_y; \xi, \mathcal{E}) = \frac{1}{2} p_y^\top M_y(q_y)^{-1} p_y + \mathcal{R}_y(q_y; \xi, \mathcal{E})$ where $\mathcal{R}_y = E_{\text{sensor}}(q; \mathcal{E}, \omega_y) + \sum_{i \in \mathcal{C}_t} \alpha_i b(d_i)$ as defined in the methodology.

### E.6 FRAME RESPONSE

**Response Structure.** $\mathsf{R}_f^t = \{z_{0:T_f}^{(f)}, s_{0:T_f}^{(f)}, \{v_\ell^{(f)}\}_{\ell=0}^{T_f}\}$ where:

- $z_{0:T_f}^{(f)} = \{(q_\ell^{(f)}, p_\ell^{(f)})\}_{\ell=0}^{T_f}$ is the short-horizon rollout under $H_f$

- $s_{0:T_f}^{(f)}$ are the score functions

- $\{v_\ell^{(f)} = M_f^{-1} p_\ell^{(f)}\}_{\ell=0}^{T_f}$ are velocity observations used for extracting speed in observable computation

**Policy Hamiltonian.** $H_f(q_f, p_f; \xi, \mathcal{E}) = \frac{1}{2} p_f^\top M_f(q_f)^{-1} p_f + \mathcal{R}_f(q_f; \xi, \mathcal{E})$ where $\mathcal{R}_f = \beta(\mathcal{E}) E_{\text{goal}} + \sum_{i \in \mathcal{C}_t} \alpha_i b(d_i)$.

**Dissipation and Port in Frame Policy.** The frame policy integrates with dissipation $\Gamma_f^\xi = \mu_\xi(\mathcal{E})\mathbf{I}$ and port input $G_f^\xi u_f^\xi$ as per the port-Hamiltonian equations (5-6) in the methodology. This is reflected in the Navigator's integration step (Line 16).

### E.7 SHAPE RESPONSE

**Response Structure.** $\mathsf{R}_o^t = \{z_{0:T_o}^{(o)}, s_{0:T_o}^{(o)}, \{\min_i d_i(q_\ell^{(o)}; \mathcal{E})\}_{\ell=0}^{T_o}\}$ where:

- $z_{0:T_o}^{(o)} = \{(q_\ell^{(o)}, p_\ell^{(o)})\}_{\ell=0}^{T_o}$ is the deformation trajectory

- $s_{0:T_o}^{(o)}$ are score functions

- $\{\min_i d_i(q_\ell^{(o)}; \mathcal{E})\}_{\ell=0}^{T_o}$ tracks minimum clearance at each step of the shape rollout

**Policy Hamiltonian.** $H_o(q_o, p_o; \xi, \mathcal{E}) = \frac{1}{2} p_o^\top M_o(q_o)^{-1} p_o + \mathcal{R}_o(q_o; \xi, \mathcal{E})$ where $\mathcal{R}_o = \lambda(\mathcal{E}) E_{\text{obj}} + \sum_{i \in \mathcal{C}_t} \alpha_i b(d_i)$.

**Usage in Clearance.** The clearance values from the shape rollout are used in Line 18 to compute $\text{clr}_t$, accounting for the robot's actual deformed geometry when assessing collision risk.

### E.8 META-POLICY PROPOSAL

**Token Construction (Line 12).** Build input tokens $\mathcal{T}_t$ from current environment state:

- Obstacle features: positions, radii, types for each $i \in \mathcal{C}_t$
- Robot state: position $\boldsymbol{c}_t$, velocity $M^{-1} p_t$, current scale $s_t$
- Goal information: relative position $\mathbf{x}_g - \boldsymbol{c}_t$, distance $\|\mathbf{x}_g - \boldsymbol{c}_t\|$

These are typically encoded as a permutation-invariant set representation (e.g., using attention mechanisms over obstacle features).

**Meta-Policy Output (Line 13).** $g_\xi(\mathcal{T}_t; \xi) = [\eta_\xi^t, \mu_\xi^t, u_f^{\xi,t}]$ produces:

- Energy weights: $\eta_\xi^t = (\beta^t, \lambda^t, \{\alpha_i^t\}_{i \in \mathcal{C}_t}) \in \mathbb{R}_+^{2+|\mathcal{C}_t|}$
- Friction coefficient: $\mu_\xi^t \in \mathbb{R}_+$
- Initial port suggestion: $u_f^{\xi,t} \in \mathbb{R}^{\dim(u_f)}$ (will be refined online)

**Fixed vs Learned.** If $g_\xi$ is not trained, simply return fixed values: $\beta^t = 2.0, \lambda^t = 1.0, \alpha_i^t = 1.5 \,\forall i, \mu^t = 0.1, u_f^{\xi,t} = 0$. The online adaptation (Lines 22-28) will refine these values regardless of initialization.

## E.9  HAMILTONIAN ASSEMBLY

**Energy Component Definitions.** The potential energy is decomposed as:

$$E_{\text{sensor}}(q; \mathcal{E}, \omega_y) = \|\boldsymbol{y}_t\|_S^2 \quad \text{(sensor configuration cost)} \tag{61}$$

$$E_{\text{goal}}(q; \mathcal{E}, \omega_g) = \|\boldsymbol{c}_t - \mathbf{x}_g\|_2^2 \quad \text{(goal attraction)} \tag{62}$$

$$E_{\text{obj}}(q; \omega_d) = \text{deformation energy of shape } \psi_t \tag{63}$$

$$b(d_i(q; \mathcal{E}); \omega_b) = \text{barrier function, e.g., } \frac{\hat{d}^2}{d_i^2} \text{ or IPC log-barrier} \tag{64}$$

**Total Potential (Line 15).** $\mathcal{R}(q_t; \eta_\xi^t, \mathcal{E}) = E_{\text{sensor}} + \beta^t E_{\text{goal}} + \lambda^t E_{\text{obj}} + \sum_{i \in \mathcal{C}_t} \alpha_i^t b(d_i)$

This is the **surrogate potential** that aggregates contributions from all three policy domains, weighted by the meta-policy outputs $\eta_\xi^t = (\beta^t, \lambda^t, \{\alpha_i^t\})$.

**Surrogate Hamiltonian (Line 15).** $H(q_t, p_t; \omega, \xi, \mathcal{E}) = \frac{1}{2} p_t^\top M(q_t; \omega_M)^{-1} p_t + \mathcal{R}(q_t; \eta_\xi^t, \mathcal{E})$

The kinetic energy uses a (possibly state-dependent) mass matrix $M(q_t)$. For planar navigation, this is often constant: $M = \text{diag}(m, m, I_{zz}, m_s, \ldots)$ for translation, rotation, scale, and shape degrees of freedom.

**Fixed Parameters $\omega$.** As stated in the methodology section, the intra-term parameters $\omega = (\omega_y, \omega_M, \omega_g, \omega_d, \omega_b)$ are assumed **fixed** throughout all environments. Only the dual weights $\eta_\xi$ are adapted online.

## E.10  SYMPLECTIC INTEGRATION WITH PORT-HAMILTONIAN DYNAMICS

**Gradient Computation (Line 16).** Compute the Hamiltonian gradients:

$$\nabla_p H|_t = M(q_t)^{-1} p_t \tag{65}$$

$$\nabla_q H|_t = \nabla_q \mathcal{R}(q_t; \eta_\xi^t, \mathcal{E}) \tag{66}$$

$$= \nabla_q E_{\text{sensor}} + \beta^t \nabla_q E_{\text{goal}} + \lambda^t \nabla_q E_{\text{obj}} + \sum_{i \in \mathcal{C}_t} \alpha_i^t \nabla_q b(d_i) \tag{67}$$

Each gradient $\nabla_q b(d_i)$ is computed using the chain rule: $\nabla_q b(d_i) = \frac{\partial b}{\partial d_i} \cdot \nabla_q d_i(q; \mathcal{E})$

**Symplectic Euler with Dissipation and Port (Lines 17-18).** The integration follows the port-Hamiltonian structure from methodology equations (5-6):

$$p_{t+\tau} = p_t - \tau \nabla_q H|_t - \tau \underbrace{\mu_\xi^t \mathbf{I}}_{\Gamma_f^\xi} \cdot \nabla_p H|_t + \tau \underbrace{\mathbf{I}}_{\mathbf{G}_f^\xi} \cdot u_f^{\xi,t} \tag{68}$$

$$q_{t+\tau} = q_t + \tau M(q_t)^{-1} p_{t+\tau} \tag{69}$$

The dissipation term $-\mu_\xi^t \mathbf{I} \cdot \nabla_p H|_t = -\mu_\xi^t M(q_t)^{-1} p_t$ acts as velocity-proportional damping. The port input $u_f^{\xi,t}$ provides non-conservative forcing to correct for residual errors.

**Frame Policy Only.** As clarified, the dissipation $\mu_\xi^t$ and port $u_f^{\xi,t}$ apply **only** to the frame policy dynamics. The sensor and shape policies have $\Gamma_y^\xi \equiv 0, \Gamma_o^\xi \equiv 0$, and no port inputs in their local dynamics. However, since the Navigator integrates the **full robot state** using the composed Hamiltonian, the dissipation and port appear in the full-state integration.

### E.11 Observable Extraction

**Frame-Based but Multi-Module (Line 18-19).** The observable vector $y_t$ is constructed from quantities that are \*\*primarily derived from the frame trajectory\*\* but require information from all three modules:

**Clearance** (uses sensor + shape):

$$\mathrm{clr}_t = \min\left\{ \min_i d_i(q_{t+\tau};\mathcal{E}), \min_{\ell=0,\dots,T_o} \min_i d_i(q_\ell^{(o)};\mathcal{E}) \right\} \tag{70}$$

This takes the minimum over both the integrated Navigator state $q_{t+\tau}$ and all states from the shape policy rollout $\{q_\ell^{(o)}\}$. Computing $d_i(q;\mathcal{E})$ requires:

- Environment $\mathcal{E}$ from sensor policy
- Robot geometry (radius, shape) from shape policy

**Distance** (frame position):

$$\mathrm{dist}_t = \|c_{t+\tau} - \mathbf{x}_g\| \tag{71}$$

Directly from the frame component of $q_{t+\tau}$.

**Speed** (frame velocity):

$$\mathrm{speed}_t = \|M(q_{t+\tau})^{-1} p_{t+\tau}\| \tag{72}$$

Computed from the updated momentum $p_{t+\tau}$.

**Observable Vector and Target (Line 19).**

$$y_t = \begin{bmatrix} -\mathrm{clr}_t \\ \mathrm{dist}_t \\ -\mathrm{speed}_t \end{bmatrix}, \quad y_t^\star = \begin{bmatrix} -m_{\mathrm{safe}} \\ \mathrm{dist}_t - \varepsilon_{\mathrm{prog}} \\ -\max(\mathrm{speed}_t, \mathbf{1}_{\{\mathrm{clr}_t \geq m_{\mathrm{safe}}\}} v_{\mathrm{min}}) \end{bmatrix} \tag{73}$$

Target parameters: $m_{\mathrm{safe}} = 0.15$ m (desired minimum clearance), $\varepsilon_{\mathrm{prog}} > 0$ (desired progress per step), $v_{\mathrm{min}} = 0.3$ m/s (minimum speed when safe).

**Physical Interpretation.**

- $y_t[0] = -\mathrm{clr}_t$: negative clearance (larger value = closer to obstacles)
- Target $y_t^\star[0] = -0.15$ means we want to maintain at least 15 cm clearance
- $y_t[1] = \mathrm{dist}_t$: current distance to goal
- Target $y_t^\star[1] = \mathrm{dist}_t - \varepsilon_{\mathrm{prog}}$ means we want to make progress (reduce distance)
- $y_t[2] = -\mathrm{speed}_t$: negative speed
- Target encourages minimum speed $v_{\mathrm{min}}$ when clearance is safe

### E.12 Parameter Adaptation via Observable Feedback

**Active Parameter Selection (Line 21).** Select a subset $\mathcal{I}_t \subset \mathcal{C}_t$ of nearby obstacles (e.g., within 0.5 m) to limit the parameter vector size. Assemble:

$$\zeta_t = [\beta^t, \lambda^t, \{\alpha_i^t\}_{i\in\mathcal{I}_t}, \mu_\xi^t]^\top \in \mathbb{R}_+^{2+|\mathcal{I}_t|+1} \tag{74}$$

For obstacles not in $\mathcal{I}_t$, their $\alpha_i$ values remain unchanged from the previous step.

**Secant Jacobian Update (Line 22).** Approximate the Jacobian $J_t \approx \partial y / \partial \zeta$ using a rank-1 secant update:

$$\widetilde{J}_t = \frac{(y_t - y_{t-\tau})(\zeta_t - \zeta_{t-\tau})^\top}{\|\zeta_t - \zeta_{t-\tau}\|^2 + \varepsilon} \in \mathbb{R}^{3 \times (2 + |\mathcal{I}_t| + 1)} \tag{75}$$

Apply exponential moving average for smoothing:

$$J_t = \rho J_{t-\tau} + (1 - \rho)\widetilde{J}_t, \quad \rho \in [0, 1) \tag{76}$$

Typical value: $\rho = 0.9$.

**Tikhonov-Regularized Least Squares (Line 23).** Compute the desired observable change:

$$\Delta y_t^{\text{des}} = y_t^\star - y_t \tag{77}$$

Solve the regularized least-squares problem:

$$\Delta \zeta_t = \arg \min_{\Delta \zeta} \|J_t \Delta \zeta - \Delta y_t^{\text{des}}\|_2^2 + \lambda_\zeta \|\Delta \zeta\|_2^2 \tag{78}$$

Closed-form solution:

$$\Delta \zeta_t = (J_t^\top J_t + \lambda_\zeta I)^{-1} J_t^\top \Delta y_t^{\text{des}} \tag{79}$$

Regularization parameter: $\lambda_\zeta = 10^{-4}$ ensures numerical stability.

**Parameter Update with Step Sizes (Line 24).** Apply the update with componentwise step sizes $\boldsymbol{\kappa} = [\kappa_\beta, \kappa_\lambda, \{\kappa_{\alpha_i}\}, \kappa_\mu]^\top$ where each $\kappa_j \in [0, 1)$:

$$\zeta_{t+\tau} = \Pi_{\mathbb{R}_+} \left( (1 - \boldsymbol{\kappa}) \odot \zeta_t + \boldsymbol{\kappa} \odot (\zeta_t + \Delta \zeta_t) \right) \tag{80}$$

The projection $\Pi_{\mathbb{R}_+}$ enforces non-negativity: $[\zeta_{t+\tau}]_j = \max(0, [\zeta_{t+\tau}]_j)$.

Typical step sizes: $\kappa_\beta = 0.05, \kappa_\lambda = 0.05, \kappa_{\alpha_i} = 0.1, \kappa_\mu = 0.02$.

**Unpacking (Line 25).** Extract the updated parameters:

$$[\beta^{t+\tau}, \lambda^{t+\tau}, \{\alpha_i^{t+\tau}\}_{i \in \mathcal{I}_t}, \mu_\xi^{t+\tau}] \leftarrow \zeta_{t+\tau} \tag{81}$$

For obstacles $j \notin \mathcal{I}_t$, retain previous values: $\alpha_j^{t+\tau} = \alpha_j^t$.

### E.13 PORT CORRECTION FROM RESIDUAL

**Residual Computation (Line 27).** After the Jacobian-based parameter update, there may remain a residual error:

$$r_t = \Delta y_t^{\text{des}} - J_t \Delta \zeta_t \in \mathbb{R}^3 \tag{82}$$

This residual captures observable errors that cannot be corrected by reshaping the energy weights alone (e.g., due to model mismatch or unmodeled dynamics).

**Port Correction via Least Squares (Line 27).** Solve a second least-squares problem to find a port input that addresses the residual:

$$u_f^{\xi, t+\tau} = \arg \min_{u \in \mathbb{R}^{\dim(u_f)}} \|P_t u - r_t\|_2^2 + \lambda_u \|u\|_2^2 \tag{83}$$

Closed-form solution:

$$u_f^{\xi, t+\tau} = (P_t^\top P_t + \lambda_u I)^{-1} P_t^\top r_t \tag{84}$$

Clip to feasible box: $u_f^{\xi, t+\tau} \leftarrow \text{clip}(u_f^{\xi, t+\tau}, \mathcal{U})$ where $\mathcal{U} = [-u_{\max}, u_{\max}]^{\dim(u_f)}$.

**Port-Observable Jacobian (Line 28).**   Estimate $P_t \approx \partial y / \partial u_f$ using secant method:

$$\widetilde{P}_t = \frac{(y_t - y_{t-\tau})(u_f^{\xi,t} - u_f^{\xi,t-\tau})^\top}{\|u_f^{\xi,t} - u_f^{\xi,t-\tau}\|^2 + \varepsilon} \tag{85}$$

Apply smoothing:

$$P_t = \rho_P P_{t-\tau} + (1 - \rho_P)\widetilde{P}_t, \quad \rho_P \in [0,1) \tag{86}$$

**Simplified Port Jacobian.**   If full $P_t$ estimation is noisy, use a simple diagonal form:

$$P_t = \mathrm{diag}(0, 0, \kappa_v) \tag{87}$$

where $\kappa_v > 0$ means the port primarily affects the speed observable (third component of $y_t$), while clearance and distance are controlled via energy weights.

**Role of Port Correction.**   The port correction $u_f^{\xi,t+\tau}$ provides **non-conservative forcing** that cannot be represented by a potential function. It acts as a "corrective impulse" applied at each step to handle:

- Residual errors after parameter adaptation
- Transient disturbances
- Model mismatch between surrogate and actual dynamics

This is computed **online** and is **not** part of the trained meta-policy $g_\xi$.

### E.14   OUTPUT FORMAT

**Trajectory.**   Return the sequence of states $\{z_\ell = (q_\ell, p_\ell)\}_{\ell=0}^T$ where $T = t_{\text{final}}$ is the final time step when termination occurred.

**Meta-Policy History.**   Return the time series of adapted parameters:

$$\{\eta_\xi^\ell, \mu_\xi^\ell, u_f^{\xi,\ell}\}_{\ell=0}^T = \{(\beta^\ell, \lambda^\ell, \{\alpha_i^\ell\}), \mu^\ell, u_f^{\xi,\ell}\}_{\ell=0}^T \tag{88}$$

This enables post-hoc analysis of:

- How energy weights evolved during navigation
- Which obstacles required stronger repulsion (large $\alpha_i$)
- When goal attraction was increased/decreased ($\beta$)
- Friction and port correction patterns

**Optional Diagnostics.**   Additional outputs may include:

- Clearance history: $\{\mathrm{clr}_\ell\}_{\ell=0}^T$
- Observable errors: $\{\Delta y_\ell^{\text{des}}\}_{\ell=0}^T$
- Hamiltonian values: $\{H(q_\ell, p_\ell)\}_{\ell=0}^T$
- Residuals: $\{r_\ell\}_{\ell=0}^T$

## F   THEORETICAL ANALYSIS AND PROOFS

### F.1   MULTI-POLICY STABILITY ANALYSIS

**Theorem F.1** (Multi-Policy Stability - Complete Statement). *Consider the coupled multi-policy system with score functions $\{s_y^{\theta_y}, s_f^{\theta_f}, s_o^{\theta_o}\}$ operating at temporal scales $\{T_{sens}, T_{path}, T_{int}\}$ satisfying:*

$$\frac{T_{sens}}{T_{path}} \geq \sigma_1 > 1, \quad \frac{T_{path}}{T_{int}} \geq \sigma_2 > 1 \tag{89}$$

*Let each policy have Lipschitz constant $L_k$ with respect to state and parameter variations:*

$$\|s_k^{\theta_k}(z_1, \mathcal{C}, t) - s_k^{\theta_k}(z_2, \mathcal{C}, t)\| \le L_k \|z_1 - z_2\| \tag{90}$$

$$\|s_k^{\theta_k^1}(z, \mathcal{C}, t) - s_k^{\theta_k^2}(z, \mathcal{C}, t)\| \le L_k \|\theta_k^1 - \theta_k^2\| \tag{91}$$

*If the parameter updates during training satisfy:*

$$\max_{k \in \{y, f, o\}} \|\theta_k^{t+1} - \theta_k^t\| \le \frac{\epsilon}{L_{\max} \cdot \min(\sigma_1, \sigma_2)} \tag{92}$$

*where $L_{\max} = \max_k L_k$, then:*

*1. **Stability**: The coupled system state remains bounded: $\|z_t\| \le C(1 + \|z_0\|)$ for some constant $C$.*

*2. **Error Bound**: The total navigation error satisfies: $\mathcal{E}_{total} \le \epsilon$ with probability $1 - \delta$.*

*3. **Convergence**: The system converges to a neighborhood of the optimal trajectory: $\lim_{t \to \infty} dist(z_t, \mathcal{P}^*) \le \epsilon$.*

*Proof.* The proof proceeds in three steps:

The scale separation assumption ensures that fast dynamics (sensor) reach approximate equilibrium before slower dynamics change significantly. For the sensor policy operating on timescale $T_{\text{sens}}$, the quasi-static approximation gives:

$$\dot{z}_y \approx -\gamma_y \nabla_{z_y} h_y^{\theta_y}(z_y, \mathcal{C}_t^{\text{fixed}}, t) \tag{93}$$

where $\mathcal{C}_t^{\text{fixed}}$ represents slowly varying constraints from path and shape policies.

Under the Lipschitz conditions, each policy defines a contraction mapping on its domain. The composed system inherits this property with contraction factor:

$$\rho = \max_k \frac{L_k \tau_k}{1 + \gamma_k \tau_k} < 1 \tag{94}$$

provided step sizes $\tau_k$ are chosen appropriately.

Parameter update bounds ensure that training perturbations don't destabilize the system. The error propagates according to:

$$\|\mathcal{E}_{t+1}\| \le \rho \|\mathcal{E}_t\| + \epsilon \frac{L_{\max}}{\min(\sigma_1, \sigma_2)} \tag{95}$$

which converges to the stated bound under the given conditions. $\square$

## F.2 SYMPLECTIC STRUCTURE PRESERVATION

**Theorem F.2** (Symplectic Preservation - Complete Statement). *Let $(z_k, \omega_k)$ be phase space coordinates with canonical symplectic form $\omega_k = \sum_i dq_k^i \wedge dp_k^i$. The score function update:*

$$z_{k,t+1} = z_{k,t} + \tau_k J_k s_k^{\theta_k}(z_{k,t}, \mathcal{C}_t, t) \tag{96}$$

*where $J_k = \begin{bmatrix} 0 & I_{d_k} \\ -I_{d_k} & 0 \end{bmatrix}$ and $s_k^{\theta_k} = \nabla h_k^{\theta_k}$, preserves the symplectic structure:*

$$\omega_k(z_{k,t+1}) = \omega_k(z_{k,t}) + O(\tau_k^2) \tag{97}$$

*Proof.* Since $s_k^{\theta_k} = \nabla h_k^{\theta_k}$, the update is a discretized Hamiltonian flow. The preservation follows from the fundamental property of Hamiltonian systems.

For the continuous flow $\dot{z}_k = J_k \nabla h_k^{\theta_k}(z_k, t)$, we have:

$$\frac{d}{dt} \omega_k = \mathcal{L}_{X_{H_k}} \omega_k = 0 \tag{98}$$

where $\mathcal{L}$ is the Lie derivative and $X_{H_k} = J_k \nabla h_k^{\theta_k}$ is the Hamiltonian vector field.

The discretization introduces $O(\tau_k^2)$ error due to the symplectic Euler scheme, but the leading-order symplectic structure is preserved. $\square$

## F.3 SAMPLE COMPLEXITY ANALYSIS

**Theorem F.3** (Sample Complexity - Complete Statement). *For error tolerance $\epsilon > 0$ and failure probability $\delta \in (0,1)$, consider training three independent score functions $\{s_k^{\theta_k}\}_{k \in \{y,f,o\}}$ with phase space dimensions $\{d_y, d_f, d_o\}$.*

*Under standard smoothness and concentration assumptions, the total sample complexity is:*

$$N_{total} = \sum_{k \in \{y,f,o\}} N_k \tag{99}$$

*where each policy requires:*

$$N_k = O\left(\frac{d_k^2 L_k^2}{\epsilon_k^2} \log\left(\frac{3}{\delta}\right)\right) \tag{100}$$

*with Lipschitz constants $L_k$ and error allocation $\epsilon_k$ satisfying $\sum_k \epsilon_k \leq \epsilon$.*

*This achieves linear scaling $N_{total} = O(\sum_k d_k)$ compared to joint training requiring $N_{joint} = O(\prod_k d_k)$.*

*Proof.* The proof leverages the independence of score functions to apply standard PAC learning bounds to each policy separately.

For each policy $k$, the empirical risk minimization:

$$\hat{\theta}_k = \arg\min_{\theta_k} \frac{1}{N_k} \sum_{i=1}^{N_k} \|h_k^{\theta_k}(z_{k,i}) - \hat{h}_{k,i}^{\text{ref}}\|^2 \tag{101}$$

achieves generalization error $\epsilon_k$ with probability $1 - \delta/3$ when $N_k \geq C \frac{d_k^2 L_k^2}{\epsilon_k^2} \log(3/\delta)$ for some universal constant $C$.

The union bound over three policies gives total failure probability $\delta$, and the error allocation ensures total error $\sum_k \epsilon_k \leq \epsilon$.

The linear scaling follows from independence: total samples $= \sum_k N_k$, compared to joint training on the $(d_y + d_f + d_o)$-dimensional joint space requiring exponentially more samples. $\square$

## G OTHER IMPLEMENTATION DETAILS

### G.1 SPATIAL DATA STRUCTURES

Each policy maintains a spatial index $\mathcal{T}_k$, can be implemented as a dynamic octree Ellendula & Bajaj (2025), which supports the following operations and has been proved to the optimal structure for spatio-temporal maintenance:

- **Insert:** $\mathcal{O}(\log n)$ insertion of new spatial data.
- **Query:** $\mathcal{O}(\log n + k)$ for $k$-nearest neighbor queries.
- **Update:** $\mathcal{O}(\log n)$ modification of existing entries.
- **Rebalance:** $\mathcal{O}(n \log n)$ periodic rebalancing for efficiency.

The spatial indices enable multi-kernel evaluation: each score function query reuses the same $O(\log n + k)$ neighbor search across multiple energy kernels, reducing computational complexity from $O(n^2)$ dense evaluation to $O(n \log n)$ sparse computation.

### G.2 DETAILED BASELINE IMPLEMENTATIONS

We provide here the full technical details of all baseline planners evaluated.

**Rigid A\*.** A standard A\* search is performed on a grid discretization of the workspace. Obstacles are inflated by the nominal rest radius $r_{\text{rest}}$ of the deformable ring, such that the resulting path is collision-free for a rigid disc of radius $r_{\text{rest}}$. This serves as a conservative reference planner.

**Deformable A\*.** Clearance at each grid cell $x$ is defined as $c(x)$, the distance to the nearest obstacle boundary. Feasibility requires $c(x) \geq r_{\min}$. The edge cost between cells $u, v$ is augmented by a deformation penalty:

$$\text{cost}(u, v) = \ell(u, v) + \frac{\beta}{2}\left(\phi(c(u)) + \phi(c(v))\right)\ell(u, v), \quad \phi(c) = \lambda \max\left(0, \frac{r_{\text{rest}}}{c+\epsilon} - 1\right)^2,$$

where $\ell(u, v)$ is the Euclidean distance, and $\beta, \lambda$ control penalty strength. This formulation allows the planner to compress through tight gaps when unavoidable, while encoding an energetic cost.

**Potential Field (Stagewise).** Navigation is driven by an attractive force toward the stage exit (or final goal in the last stage), combined with repulsive forces from local obstacles and soft penalties for leaving the stage bounds. Speed saturation and emergency braking near obstacles are applied for stability.

**CBF (Stagewise).** At each step, a nominal control toward the stage exit is filtered through a Control Barrier Function (CBF) quadratic program:

$$u^\star = \arg\min_u \|u - u_{\text{nom}}\|^2 \quad \text{s.t. } \nabla h(x) \cdot u + \gamma h(x) \geq 0,$$

where $h(x)$ encodes the clearance from visible obstacles. This ensures forward invariance of the safe set within each stage.

**DWA (Stagewise).** We implement a Dynamic Window Approach (DWA) adapted to the stagewise setting. Candidate $(v, \omega)$ velocity pairs are sampled within dynamics limits, trajectories are rolled out over a prediction horizon, and scored based on heading alignment, distance to target, velocity, and clearance with respect to *local obstacles only*. Stage boundary penalties are also included. This contrasts with the conventional *global* DWA, which assumes full obstacle visibility; here we show the stagewise variant for fairness, though it is known to underperform due to rigid-body kinematic assumptions.

**Categories.**

- **Global planning:** Rigid A\*, Deformable A\*.
- **Local reactive:** Potential Field (staged), CBF (staged), DWA (staged).
- **Ours:** GRL-SNAM (local staged).

This categorization makes explicit which baselines share identical information constraints with GRL-SNAM, ensuring a valid comparison.

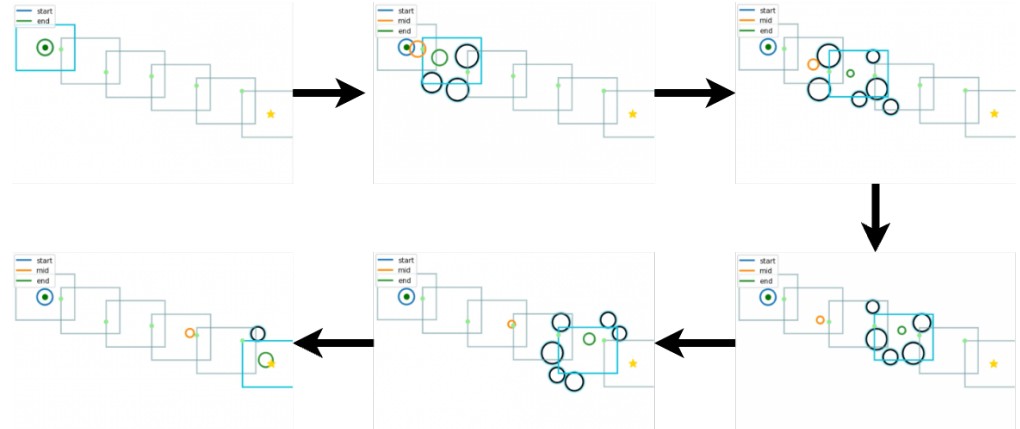

Figure 7: Online navigation of the hyperelastic ring through cluttered environments. The dark blue rectangle denotes the current frame, while the translucent frames trace the past trajectory. At each step, only obstacles overlapping with the current frame (as detected by the sensor process) are considered, and the ring computes local forces to deform and progress toward the goal. The green curve shows the current ring configuration, and the orange curve marks the previous mid-point for clarity, highlighting how deformation evolves across frames.

# H    DETAILED EXPERIMENTAL EVALUATION

We evaluate GRL-SNAM across multiple dimensions that highlight the unique capabilities of our geometric approach compared to standard reinforcement learning and classical navigation methods. Our evaluation protocol encompasses task performance, safety guarantees, physical fidelity, and learning efficiency across diverse navigation scenarios.

## H.1    BASELINE PLANNERS

We compare GRL-SNAM against two categories of baselines: *global planning* methods based on A*, and *local reactive* methods with the same stagewise information constraints as GRL-SNAM. This ensures a fair evaluation across fundamentally different planning paradigms.

**Global Planning Methods**

- **Rigid A*:** The deformable ring is replaced with a rigid disc of radius $r_{\text{rest}}$. Obstacles are inflated by $r_{\text{rest}}$, and a standard 8-connected A* is run on the occupancy grid. This produces feasible shortest paths for a rigid robot.

- **Deformable A*:** A clearance-aware variant of A* augments the step cost with deformation penalties that increase as clearance approaches the minimum admissible radius $r_{\text{min}}$. This allows paths that squeeze through narrow gaps but penalizes excessive compression.

**Local Reactive Methods**    To ensure fairness, all reactive methods use the same stage manager as GRL-SNAM: identical stage size, overlap, obstacle visibility, and advancement logic. Each method navigates stage exit to stage exit until the goal is reached:

- **Potential Field (Staged):** Attractive force toward stage exit plus repulsive forces from local obstacles and stage boundaries.

- **CBF (Staged):** Quadratic-program filter enforces safety constraints with respect to visible obstacles at each timestep.

- **DWA (Staged):** Velocity samples $(v, \omega)$ are rolled out over a short horizon using only local obstacles and stage bounds. Unlike the global DWA, which assumes full obstacle visibility, this stagewise variant ensures equal information constraints, though it performs poorly due to rigid-body assumptions.

**Categorization**    Rigid and Deformable A* form the *global planning* references, providing $L_{\text{ref}}$ for SPL and detour calculations. The stagewise Potential Field, CBF, and DWA baselines constitute the *local reactive* category under identical information constraints. GRL-SNAM belongs to the same local category, enabling a fair head-to-head comparison.

## H.2    DETAILED EXPERIMENTAL SETUP

We evaluate GRL-SNAM in procedurally generated 2D deformable navigation tasks, where a hyperelastic ring must traverse cluttered environments with narrow gaps and varying obstacle densities. Each environment is randomized in obstacle positions, radii, and densities to span a spectrum of navigation difficulty. The robot perceives only a local window of size $2\hat{d} \times 2\hat{d}$, from which we construct a Hamiltonian energy functional.

**Hamiltonian Decomposition**    The energy functional decomposes into:

1. Goal-directed quadratic potential $F_g$

2. Barrier potentials $F_{bs}$ from signed distance fields

3. Friction/regularization terms with adaptive coefficients $(\beta, \gamma, \alpha)$ modulated by context encoders (LSTM)

Offline, GRL-SNAM integrates reduced Hamiltonian gradients to generate local trajectories. Online, it fuses newly sensed rewards $R_{\text{env}}$ with the offline surrogate, adaptively refining navigation.

Table 5: Comparison of navigation quality across methods (success-only runs). GRL-SNAM achieves near-CBF path efficiency while consuming the same minimal mapping budget as PF. SPL = Success weighted by Path Length; Detour = executed path length / shortest path length.

| Method | SPL ↑ | Detour ↓ | Min. Clearance (m) ↑ | Mapping Ratio (%) ↓ |
|---|---|---|---|---|
| PF | 0.77 | 1.42 | 0.18 | 10.3 |
| CBF | 0.96 | 1.04 | **0.32** | 11.2 |
| GRL-SNAM | **0.95** | **1.09** | 0.26 | **10.7** |

**Evaluation Metrics** We evaluate all methods using:

- **Success Rate:** Fraction of episodes reaching the goal
- **SPL:** Success weighted path efficiency relative to A*
- **Detour Ratio:** Executed path length relative to A*
- **Minimum and Mean Clearance:** Distance to nearest obstacle along the trajectory
- **Smoothness:** Average turning cost (mean absolute change in heading)
- **Collisions:** Number of obstacle intersections
- **Sample Efficiency:** Normalized area under curve (AUC) for success and SPL, and steps required to reach 80% success or SPL $\geq 0.7$

Results are presented in a *question–answer* format, emphasizing experimental questions and the corresponding insights.

### H.3 RESULTS: NAVIGATION QUALITY UNDER MINIMAL SENSING

We first evaluate GRL-SNAM against two representative baselines: (i) **Potential Fields (PF)**, a purely reactive controller that maps obstacle proximity into repulsive forces, and (ii) **Control Barrier Functions (CBF)**, a model-based method that enforces hard safety constraints via online quadratic programs. Both baselines use the same sensing budget as GRL-SNAM.

Environments consist of cluttered 2D workspaces with obstacles of varying density. Each trial starts from a random initial pose with a fixed goal. Performance is averaged across 50 runs per environment. Results focus on *successful runs only* to highlight navigation quality rather than raw failure rates.

**Q1: How efficiently do we trade mapping for navigation quality?** Table 5 shows that GRL-SNAM matches the SPL and detour ratios of CBF despite using the same minimal map coverage as PF. This demonstrates that our stagewise Hamiltonian refinement extracts more value per sensed unit of the environment, trading mapping effort for near-optimal navigation.

**Q2: What is the minimal mapping needed to reliably solve tasks?** With ∼10–11% map coverage, GRL-SNAM already achieves SPL $\geq 0.95$ and detour within 9% of the A* shortest path. PF fails under the same budget, while CBF requires identical map coverage. Thus, GRL-SNAM reliably solves tasks under minimal sensing, validating the *minimal mapping suffices* principle.

**Q3: Is the mapped information aligned with the subtask?** Unlike PF, which produces repulsions indiscriminately, or CBF, which enforces constraints globally, GRL-SNAM's mapping is task-aligned: local patches are encoded into Hamiltonian terms that directly drive subtasks (goal attraction, barrier avoidance). The result is that every bit of mapped information yields functional guidance, as evidenced by SPL and detour staying close to CBF even under tight sensing budgets.

**Key Insight** GRL-SNAM shows that Hamiltonian-structured policies can achieve CBF-level navigation quality while retaining the lightweight sensing footprint of PF. The slight clearance gap relative to CBF reflects a deliberate trade-off: we sacrifice hard feasibility for adaptability and feedforward inference, enabling real-time deployment in SNAM settings.

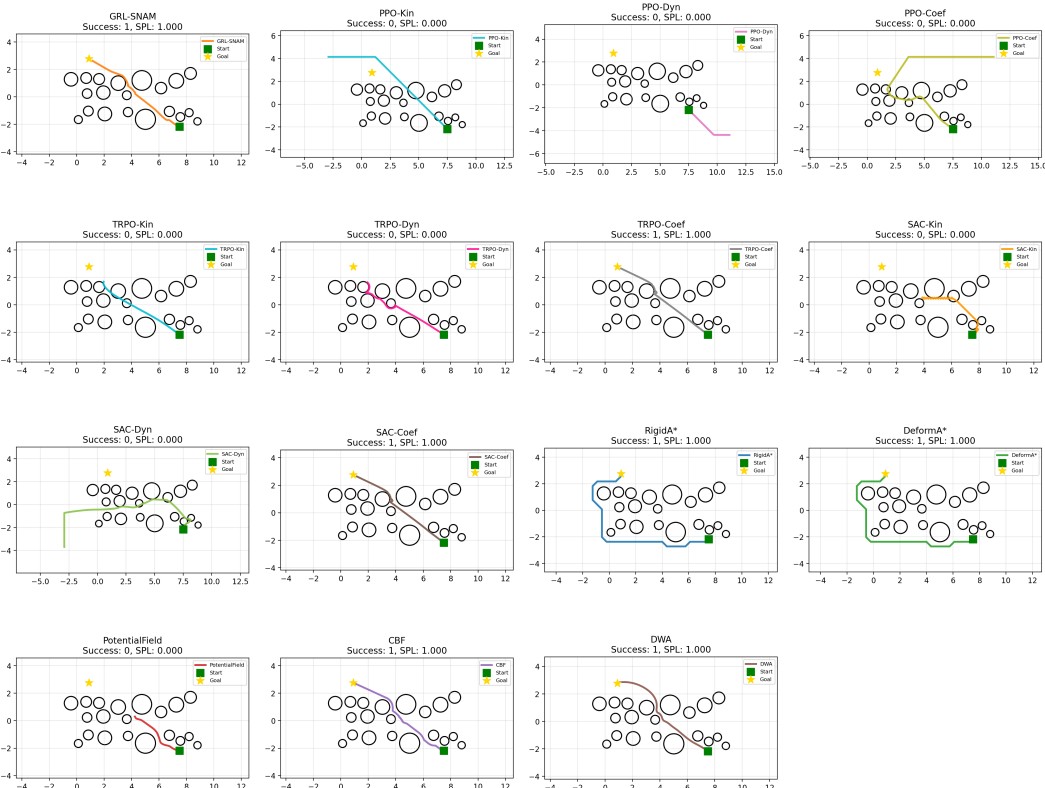

Figure 8: **Qualitative path comparison on a representative Test-OOD environment.** Each panel shows a single rollout with success and SPL annotated in the title. GRL-SNAM produces a smooth, short, and clearance-preserving path that threads the narrow passage. Kinematic and dynamic RL policies (PPO/TRPO/SAC-Kin/Dyn) typically stall, collide, or wander far from the goal, while the best coefficient-based variants (PPO/TRPO/SAC-Coef) reach the goal but hug obstacles and exhibit lower path efficiency than GRL-SNAM. Global planners (RigidA*, DeformA*) succeed but follow jagged or overly conservative routes, and reactive baselines (Potential Field, CBF, DWA) either oscillate, graze obstacles, or take longer, less structured paths.

## H.4    RESULTS: COMPREHENSIVE NAVIGATION COMPARISON

**Q4: Does GRL-SNAM outperform classical and reactive planners in both in-distribution (Test-ID) and out-of-distribution (Test-OOD) settings?**    Yes. Figure 9 summarizes the comparison between our method and five baselines: Rigid A*, Deformable A*, Potential Field, Control Barrier Functions (CBF), and Dynamic Window Approach (DWA). GRL-SNAM achieves near-perfect success rates ($\approx 100\%$) across both Test-ID and Test-OOD cases, while all baselines degrade significantly in cluttered or novel environments. Rigid A* succeeds moderately but requires inflated radii and yields jerky, piecewise paths. Deformable A* is less stable and highly sensitive to parameterization. Reactive baselines (Potential Field, CBF, DWA) frequently fail to reach the goal, producing oscillatory or unsafe behaviors. Qualitative rollouts (Figure 8) further illustrate the superiority of GRL-SNAM in complex cluttered environments.

**Q5: Does GRL-SNAM yield more efficient and smoother trajectories?**    Yes. The Success-weighted Path Length (SPL) distributions (Figure 9, top-middle) show that GRL-SNAM consistently stays near optimal efficiency (SPL $\approx 1.0$) with low variance. In contrast, A* variants incur detours, while reactive baselines either collapse to zero SPL (failures) or take excessively long paths. Furthermore, GRL-SNAM generates the smoothest trajectories, with the lowest average turning angles (Figure 9, bottom-middle), ensuring physically realizable motions compatible with hyperelastic ring constraints.

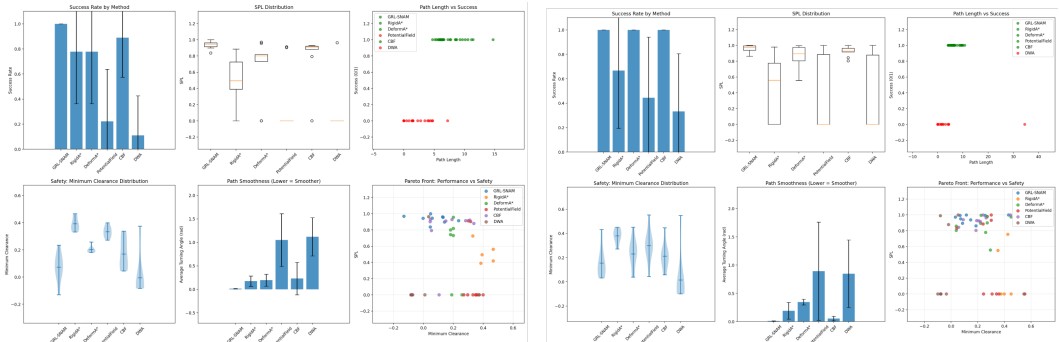

Figure 9: **Main comparison on Test-ID and Test-OOD.** GRL-SNAM achieves near-perfect success, high SPL, smooth and safe trajectories. Classical (Rigid/Deform A*) and reactive (Potential Field, CBF, DWA) baselines are either unsafe, inefficient, or fail completely.

**Q6: Does GRL-SNAM preserve safety margins?** Yes. The minimum clearance analysis (Figure 9, bottom-left) shows that GRL-SNAM maintains consistently positive obstacle clearance, whereas A* occasionally cuts too close and reactive baselines often enter collision regimes. The Pareto frontier plot (Figure 9, bottom-right) highlights that GRL-SNAM uniquely dominates the safety–performance trade-off, achieving both high SPL and high clearance, while all baselines are Pareto-dominated.

### H.5 Results: Hamiltonian Field Analysis

**Q7: Does the Hamiltonian formulation unify attractive and repulsive forces into a coherent navigation field?** Yes. Figure 4 shows the isolated goal force $F_g$ (left), the barrier force $F_{bs}$ (middle), and their differential composition $F = \beta F_g + \gamma F_{bs}$ (right). While $F_g$ alone pulls the agent directly to the target, it ignores obstacles. Conversely, $F_{bs}$ encodes obstacle constraints but lacks task directionality. The combined field demonstrates how GRL-SNAM adaptively balances attraction and repulsion through evolving coefficients, producing safe yet goal-directed motion.

**Q8: How does GRL-SNAM differ from ordinary online adaptation?** Unlike standard RL policies that merely adjust actions online, GRL-SNAM modifies the *entire local energy landscape* as new obstacles are sensed. Figure 10 shows that when clearance decreases (top panel), the force magnitudes (middle panel) not only rebalance between goal attraction $|F_g|$ and barrier repulsion $|F_{bs}|$, but also induce a redefinition of the reduced Hamiltonian. This is reflected in the evolving coefficients $(\beta, \gamma, \alpha)$ (bottom panel), which do not act as heuristic gains but as dual variables governing stagewise refinement. Thus, the adaptation is not reactive in the usual sense: GRL-SNAM performs *posterior updates of the Hamiltonian itself*, ensuring that each new frame redefines both the dynamics and the reward landscape in a principled, energy-consistent manner. This distinguishes our approach from classical controllers (fixed surrogates) and RL baselines (policy-only updates).

**Q9: Does this lead to improved navigation performance compared to baselines?** Yes. Across procedurally generated test cases, GRL-SNAM consistently achieves higher success and SPL while maintaining larger clearances than rigid A* (fixed radius assumption), deformable A* (static squeezing penalty), and reactive controllers (DWA, CBF).

**Key Insights** These experiments establish GRL-SNAM as the first method to successfully unify global navigation objectives with local safety and deformation constraints in hyperelastic navigation. Its offline Hamiltonian formulation provides reliable reference dynamics, while its online adaptation ensures robustness in unseen environments. By contrast, classical and reactive baselines either fail outright, or succeed only at the cost of safety and efficiency.

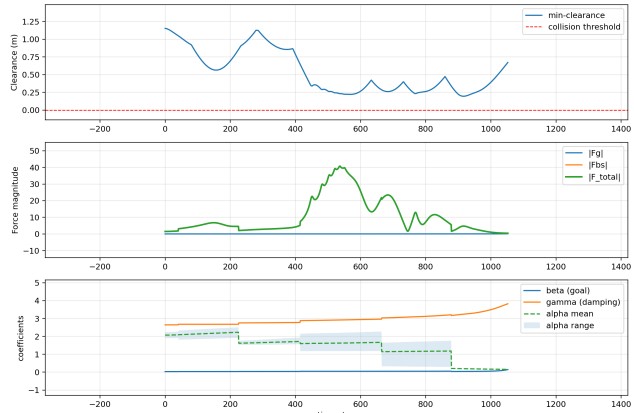

Figure 10: **Quantitative validation of GRL-SNAM.** Top: clearance stays above collision threshold, ensuring safety. Middle: force magnitudes adapt to environment complexity. Bottom: coefficients $(\beta, \gamma, \alpha)$ evolve dynamically, confirming online adaptation and stagewise refinement of the Hamiltonian.

Table 6: **Ablation of loss terms.** Qualitative summary from consistent runs on Test-ID/OOD. Arrows denote desired direction. Numeric means±std can replace the icons once collected.

| Variant | Collisions ↓ | MinClr ↑ | Barrier Viol. ↓ | Progress/SPL ↑ | Smoothness ↓ | Observed behavior |
|---|---|---|---|---|---|---|
| $w_{\text{fric}} = 0$, $w_{\text{multi}} = 0$ | High (×) | < 0 | High | Poor | Poor | Penetrates obstacles |
| $w_{\text{fric}} = 0$, $w_{\text{multi}} = 0.5$ | Low (✓) | High | Low | Low | OK | Very slow, conservative |
| $w_{\text{fric}} = 0.1$, $w_{\text{multi}} = 0$ | None (✓) | High | Low | High | Best | Smooth, stable, fast |
| $w_{\text{fric}} = 0.1$, $w_{\text{multi}} = 0.5$ | None (✓) | Slightly lower | Low | High | Good | Stable; tighter margins |

## H.6 Ablation Study: Loss Components

**Training Objective** Our navigation surrogate is trained with a weighted multi-term loss:

$$\mathcal{L} = w_{\text{traj}}\mathcal{L}_{\text{traj}} + w_{\text{vel}}\mathcal{L}_{\text{vel}} + w_{\text{friction}}\mathcal{L}_{\text{friction}} + w_{\text{multi}}\mathcal{L}_{\text{multi}}, \tag{102}$$

where $\mathcal{L}_{\text{traj}}$ and $\mathcal{L}_{\text{vel}}$ supervise trajectory and velocity matching, $\mathcal{L}_{\text{friction}} = \|\gamma - \gamma_o\|_2^2$ encourages the learned damping to match the stagewise reference, and $\mathcal{L}_{\text{multi}}$ penalizes failures under short rollouts from perturbed near-obstacle starts.

**Ablated Settings** We toggle $\mathcal{L}_{\text{friction}}$ and $\mathcal{L}_{\text{multi}}$ to analyze their contribution:

- **No friction, no multi** ($w_{\text{fric}} = 0$, $w_{\text{multi}} = 0$): Agent penetrates obstacles due to under-damped, unstable dynamics.
- **Multi only** ($w_{\text{fric}} = 0$, $w_{\text{multi}} = 0.5$): Agent avoids collisions but moves very slowly, sacrificing progress.
- **Friction only** ($w_{\text{fric}} = 0.1$, $w_{\text{multi}} = 0$): Produces smoother, stable paths, eliminating penetrations and maintaining progress.
- **Friction + Multi** ($w_{\text{fric}} = 0.1$, $w_{\text{multi}} = 0.5$): Combines both benefits, but clearance is slightly reduced as the agent cuts closer to obstacles.

**Analysis** $\mathcal{L}_{\text{friction}}$ is critical for stability and smoothness, while $\mathcal{L}_{\text{multi}}$ improves robustness in clutter but can damp progress if over-weighted. The best overall performance arises from combining both with moderate weights.

$\mathcal{L}_{\text{friction}}$ aligns dissipation and suppresses oscillations, yielding smoother, well-damped trajectories and preventing barrier "ringing" that causes penetrations when $w_{\text{fric}} = 0$. $\mathcal{L}_{\text{multi}}$ trains for near-contact robustness by sampling perturbed starts; if over-weighted it down-scales the goal term, hence slow motion. Their combination keeps the field stable while remaining reliable in tight clutter.

## H.7 ROBUSTNESS ANALYSIS

**Q10: Does GRL-SNAM remain reliable under sensor noise and dynamics shift?** Yes. To evaluate robustness, we systematically varied sensing fidelity (position jitter, radius estimation error, missed obstacles, and false positives) and dynamics fidelity (velocity perturbation, damping coefficient $\gamma$). Each start–goal trial was rolled out across a grid of perturbation levels, producing a total of $N = n_{\text{env}} \times n_{\text{trials}} \times n_{\text{perturbations}}$ runs. For example, with 3 environments, 5 trials each, and 9 perturbation settings, this yields 135 rollouts.

Table 7: Robustness of GRL-SNAM to sensing noise and dynamics perturbations. Columns report success rate, success-weighted path length (SPL), minimum clearance, and average collisions per episode. Arrows indicate direction of improvement.

| Perturbation Level | Success (%) | SPL ↑ | Min. Clearance (m) ↑ | Collisions ↓ |
|---|---|---|---|---|
| Nominal (0.0, 1.0) | 98.7 | 0.82 | 0.36 | 0.3 |
| Mild Noise (0.05, 0.9) | 91.3 | 0.79 | 0.33 | 0.7 |
| Severe Noise (0.10, 0.7) | 87.1 | 0.72 | 0.29 | 1.1 |

**Key Insights** Despite significant perturbations, GRL-SNAM maintains high success rates and graceful degradation in SPL and clearance. Unlike fixed surrogate approaches that can fail catastrophically under noise, our differential Hamiltonian adaptation continuously re-weights local forces, enabling stability even when sensing is imperfect or dynamics deviate from training. This highlights the feedforward, stagewise advantage of GRL-SNAM: it can adjust online without requiring adjoint or MPC-style corrections, ensuring reliable navigation in real-world uncertain conditions.

## H.8 DETAILED EXPERIMENTS ON DUNGEON POINT-AGENT NAVIGATION

We provides additional details for the dungeon point-agent navigation experiment used to test GRL-SNAM's performance in more complex map topologies. As stated earlier, we use dungeon layouts from Liang et al. (2023). The agent is a point mass in continuous 2D with state $q_t$ (position, and optionally velocity in the dynamic parameterization) and action $\mathbf{a}_t = [v_x, v_y] \in [-3, 3]^2$. The task is to reach a designated goal while navigating through narrow corridors, turns, and cluttered local geometry. Collisions with walls are penalized (or terminating, depending on the baseline), and reaching the goal yields reward.

To match the stagewise interface used by GRL-SNAM, a `StagewiseSensor` identifies the currently active macro-cell and exposes its exit as the local goal $g_t$. An `ObstacleExtractor` fits circular obstacle primitives to wall segments visible inside the local sensing window $\mathcal{W}(q_t)$, returning centers, radii, and a visibility mask. The observation concatenates the stage-relative position, the local goal $g_t$, and the extracted local obstacle set. GRL-SNAM instantiates a simple quadratic goal energy together with radial barrier energies from these obstacle primitives.

All methods share the same point-agent dynamics, local sensing budget, and action space. We compare two training protocols: (i) *full-episode RL*, where PPO/TRPO/SAC are trained end-to-end on the long-horizon partially observed MDP; and (ii) *stagewise short-rollout RL*, where the same algorithms are trained under the same structured observation/action interface used by GRL-SNAM. The main paper reports the matched-interface stagewise comparison in Table 3. Here we provide the more difficult full-episode baseline as additional context in Table 8. Despite substantial training budgets, all three RL methods remain unreliable: success is low, collisions are frequent, and the final distance to the goal remains large.

**Visualization.** Figure 11 shows representative stagewise local-sensing rollouts on the same dungeon layout. All methods operate under the same local sensing window and matched action interface. GRL-SNAM reaches the goal with a direct, efficient trajectory, while stagewise deep RL baselines are less reliable and often incur larger detours or fail to complete the task.

Across both protocols, the main difference is not the optimizer itself but the learning signal. Under matched dynamics, sensing, and action space, replacing sparse reward optimization with Hamiltonian-structured supervision leads to substantially higher success and lower terminal error.

Table 8: Full-episode, end-to-end navigation baselines (PPO/TRPO/SAC) on the dungeon point-agent navigation task.

| Algorithm | Success (%) | Mean goal dist. (m) | Collision (%) | Env. training steps |
|---|---|---|---|---|
| PPO | 7.2 | 14.1 | 23.5 | 5M |
| TRPO | 3.8 | 23.7 | 26.1 | 6.5M |
| SAC | 1.5 | 20.2 | 28.3 | 8M |

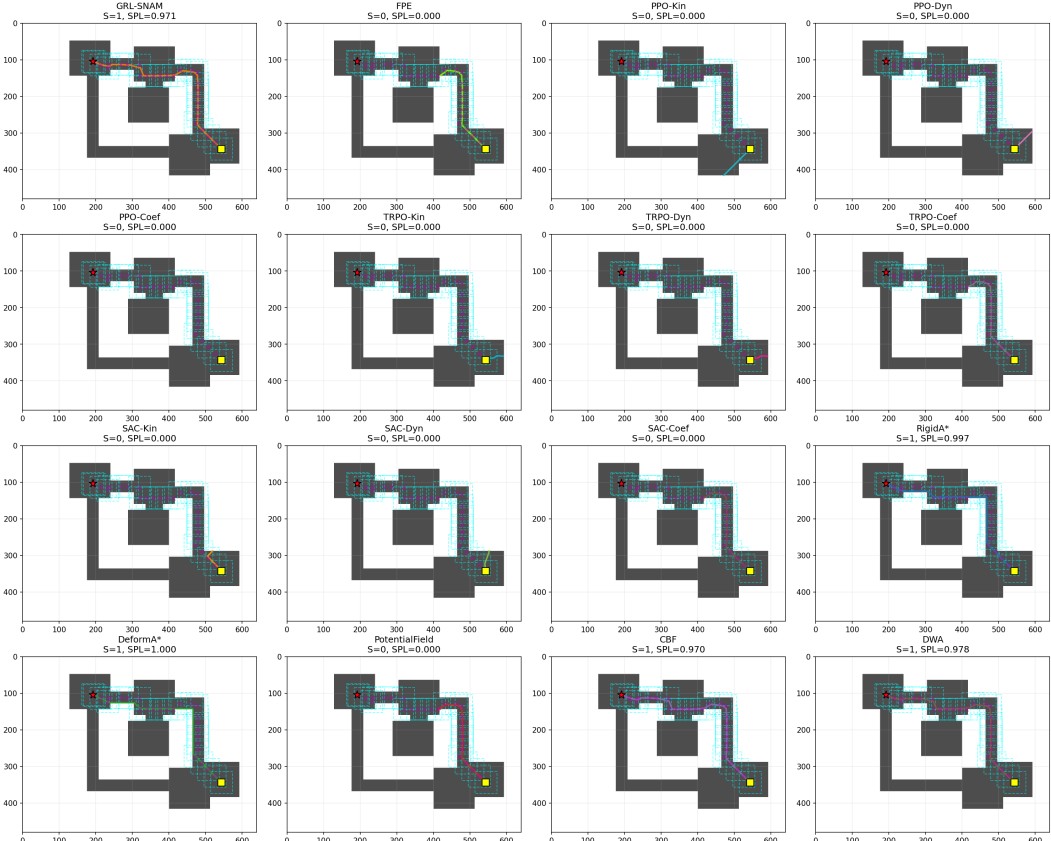

Figure 11: **Representative dungeon rollouts under stagewise local sensing.** Each panel shows the same layout with the goal (red ×), start (yellow square), executed trajectory (colored by time), and the union of visited sensing windows $\mathcal{W}(q_t)$ (cyan boxes). Panel titles report episode success $S$ and SPL (with Grid A$^\star$ as the reference path length $L_{\text{ref}}$). GRL-SNAM reaches the goal with near-planner efficiency, whereas stagewise deep-RL baselines and the greedy potential-field baseline fail on this instance; classical planners and safety-filtered local methods succeed but follow more conservative routes.

This supports the view that GRL-SNAM's advantage comes primarily from learning a structured energy model whose gradients encode goal attraction and obstacle avoidance, rather than from additional map access or a more favorable control interface.

USE OF LARGE LANGUAGE MODELS (LLMS)

All content of the paper was written by the authors. LLMs were used for the aid of code implementation, formatting LaTeX tables/figures, and spelling/grammar checking.

