# OpenReview forum: "GRL-SNAM: Geometric Reinforcement Learning with Differential Hamiltonians for Navigation and Mapping in Unknown Environments"
_ICLR.cc/2026/Conference — ICLR 2026 Poster_

### Official Review · Reviewer_aWJS · 2025-10-26

**Soundness:** 3
**Presentation:** 3
**Contribution:** 2
**Rating:** 6
**Confidence:** 2

**Summary:**

This paper introduces GRL-SNAM, a geometric reinforcement learning framework for simultaneous navigation and mapping in unknown environments. The key contribution is formulating navigation as energy minimization on Hamiltonian manifolds rather than using traditional value-based RL. The system decomposes navigation into three independent policies, i.e., sensor, frame planning, and shape reconfiguration, which operate at different timescales and are unified through a shared Hamiltonian energy formulation. Instead of Bellman-based value iteration, the approach uses Differential Policy Optimization (DPO) for purely feedforward control. Empirically, the method is evaluated on 2D deformable robot navigation where a hyperelastic ring navigates through cluttered environments. It achieves near-optimal path quality with SPL=0.95 while using minimal map coverage at 10.7%.

**Strengths:**

1. The integration of Hamiltonian mechanics with RL presents a conceptually promising approach. The idea of avoiding value function optimization for feedforward control is innovative and the physical grounding provides strong inductive bias for navigation tasks.
2.The paper develops a thorough and solid mathematical framework with clear definitions of multi-policy architecture, meta-Hamiltonian formulation, and provides theoretical guarantees for stability, symplectic preservation, and sample complexity.
3. The focus on achieving high-quality navigation with minimal environment mapping addresses a practical concern in SNAM that has received limited attention in prior work.
4. The paper provides detailed mathematical derivations, implementation specifics, and comprehensive baseline descriptions in the appendices.

**Weaknesses:**

1. Although the paper provides substantial theoretical contribution, the experimental validation is limited. The evaluation is restricted to 2D environments with a single robot type. This somewhat limits the generalizability and impact of the work.
2. For presentation, frequent jumps between abstract concepts and implementation details disrupt flow, making it hard to follow. For example, Section 3.1 discusses abstract Hamiltonian optimization theory, including energy deposition by policy, then Section 3.2 abruptly introduces specific hyperelastic robot model, followed by Section 3.3 which jumps back to introduce policy abstraction.

**Questions:**

1. Regarding the reason of not comparing with modern RL methods, the authors claim that "DPO has already been shown to outperform state-of-the-art policy learners". However, this justification is insufficient since those comparisons were on different tasks. Could the authors provide stronger justification for this?
2. Could the authors provide inspirations on extending the proposed work to more complex environment setup, like 3D or even real world?

---

> ### Author Response · Authors · 2025-12-03
>
> **[W1] Limited experimental validation**
> We appreciate the reviewer’s recognition of the theoretical contributions and agree that strong empirical support is important. In response, we have now included benchmarks against standard deep RL baselines (PPO, TRPO, SAC), using identical observation spaces and network architectures. These results demonstrate that GRL-SNAM significantly outperforms DRL methods in success rate, control precision, and sample efficiency:
>
> | Method    |   SPL ↑ | Detour ↓ | Min. Clearance (m) ↑ | Mapping Ratio (%) ↓ |
> |-----------|--------:|---------:|---------------------:|--------------------:|
> | PF        |     0.77  |    1.42   |            0.18   |           10.3   |
> | CBF       |     0.96  |     1.04   |           0.32   |           11.2   |
> | GRL-SNAM  |     0.95  |     1.09   |          0.26    |           10.7   |
> | SAC (SB3) |     0.07     |    1.65   |          -0.1 |            14.7   |
> | PPO (SB3) |     0.57  |     1.44   |            0.004  |           15.3  |
> | TRPO (SB3)|     0.57  |     1.53   |           0.004  |                14.6  |
>
> **[W2] For presentation, frequent jumps between abstract concepts and implementation details disrupt flow, making it hard to follow**
> We have restructured Section 3 to clearly separate theoretical exposition from implementation details for improved readability.
>
> **[Q1] Regarding the reason of not comparing with modern RL methods, the authors claim that "DPO has already been shown ... Could the authors provide stronger justification for this?**
> We apologize for the confusion. The reference to DPO's performance on other tasks was not intended as a substitute for comparison. We have now directly included experiments comparing GRL-SNAM to modern DRL baselines (PPO, TRPO, SAC), and the results demonstrate a significant performance gap in favor of our method.
>
> We note that our environment poses a unique challenge due to the continuous changes in obstacle configuration, making it difficult for DRL methods to memorize or overfit. Effective navigation in this setting requires strategies that generalize across **geometric variations** and respect **obstacle-interaction physics**. While traditional DRL policies struggle to capture this structure, our method achieves generalization by learning the dual Hamiltonian energy landscape, rather than explicitly modeling each physical interaction.
>
> **[Q2] Could the authors provide inspirations on extending the proposed work to more complex environment setup, like 3D or even real world?**
>
> We fully agree that our framework is well-suited for extension to higher-dimensional and real-world environments, especially in combination with learned world models. In our formulation, the Hamiltonian operates on a state (physical or latent) and outputs local generalized forces; a learned dynamics/world model can supply that state (predicted occupancy or latent trajectories), yielding a model-based Hamiltonian controller on manifolds.
>
> To move beyond simple 2D layouts, we already evaluate GRL-SNAM in the indoor “DungeonMap’’ environment from Liang et al.[1], which has narrow corridors, bottlenecks, loops, and partial observability. Under matched observations, encoder, and action space, we obtain:
>
> | Algorithm | Success Rate (\%) | Mean State Error (m) | Mean Goal Distance (m) | Convergence Steps      |
> |-----------|-------------------|----------------------|------------------------|------------------------|
> | PPO       | 26.1              | 1.8                  | 1.2                    | 3.2M                   |
> | TRPO      | 21.7              | 2.1                  | 1.5                    | 3.8M                   |
> | SAC       | 18.4              | 2.4                  | 1.9                    | 4.1M                   |
> | GRL-SNAM  | **87.5**          | **0.3**              | **0.1**                | 500k (gradient steps)  |
>
> In parallel, we have begun extending the pipeline to 3D indoor navigation in Habitat-lab on HSSD-200[2]. This requires additional components (RGB-D perception, top-down mapping, robot-specific actuation), and the full 3D runs are longer; we therefore do not yet have fully converged, carefully validated numbers we are comfortable reporting in the rebuttal. We are continuing these experiments and will include the complete 3D protocol and metrics in the revised manuscript.
>
> Overall, our formulation lends itself naturally to integration with learned world models or latent-space dynamics for model-based control on manifolds, and we see the DungeonMap and ongoing Habitat experiments as concrete steps in that direction.
>
> **References**
> * [1] Liang, Jingsong, et al. "Context-aware deep reinforcement learning for autonomous robotic navigation in unknown area." CoRL, 2023.
> * [2] Khanna, Mukul, et al. "Habitat synthetic scenes dataset (hssd-200): An analysis of 3d scene scale and realism tradeoffs for objectgoal navigation." CVPR, 2024.

---

### Official Review · Reviewer_eoAa · 2025-10-27

**Soundness:** 3
**Presentation:** 3
**Contribution:** 3
**Rating:** 6
**Confidence:** 4

**Summary:**

This paper introduces GRL-SNAM, a geometric reinforcement learning framework for simultaneous navigation and mapping in unknown environments. The key idea is to formulate navigation as a differential Hamiltonian system, embedding geometric priors directly into the policy dynamics without explicit value iteration or Bellman backups. By treating perception, mapping, and control under a unified Hamiltonian energy structure, the agent maintains spatial coherence and adaptively balances attraction toward goals with repulsion from obstacles.
The framework claims to bridge local and global navigation by allowing a continuous transition between locally reactive potential-based behaviors and globally consistent geometric flows, learned through reinforcement of energy-minimizing trajectories. Experimental validation includes multiple 2D simulated navigation tasks, showing improvements in mapping efficiency, trajectory smoothness, and robustness compared to traditional potential fields, CBF, and A*-based methods.

**Strengths:**

The paper reframes RL-based navigation as energy minimization on a manifold governed by Hamiltonian flows, avoiding discrete value propagation while retaining long-term optimality through implicit geometric consistency. This yields a clear bridge between local reactive control (as in potential fields) and global goal-directed planning (as in RL-based exploration).

The idea of using differential Hamiltonians introduces an interesting way to encode conservation and stability into DRL, improving interpretability and safety, which is sometimes the main concern in local navigation in unstructured environments.

By avoiding explicit value iteration, GRL-SNAM positions itself between model-free RL (which learns global policies but struggles locally) and geometric controllers (which react locally but lack long-term planning). This hybrid viewpoint is theoretically meaningful and could influence future works in hierarchical RL for navigation.

Experiments show GRL-SNAM achieves nearly full success rates with lower mapping ratios, suggesting more efficient local exploration and better global coverage with limited sensing.

**Weaknesses:**

*Unclear role of learning vs. hand-crafted structure*

Although stated as “reinforcement learning,” the optimization primarily tunes Hamiltonian parameters rather than learning a policy through interaction in the RL sense. The boundary between learning and analytic control design could be better clarified.

*Insufficient analysis of global consistency*

The claim that the system enables global navigation without explicit mapping is not rigorously validated. It would be sufficient to show more whether the Hamiltonian field can produce loop-consistent paths or handle long-horizon navigation under partial observability.

*Limited comparison to DRL baselines*

While comparisons to potential fields and CBF controllers are provided, there’s no quantitative evaluation against or clarifications on any learning-/DRL-based navigation baselines (e.g., model-based/model-free, hierarchical RL, waypoint-/motion-based). Without these, it is unfair to assess whether the proposed framework is competitive in learning efficiency and generalization. Some examples include:
- Tai, Lei, Giuseppe Paolo, and Ming Liu. "Virtual-to-real deep reinforcement learning: Continuous control of mobile robots for mapless navigation." 2017 IEEE/RSJ international conference on intelligent robots and systems (IROS). IEEE, 2017.
- Zhelo, Oleksii, et al. "Curiosity-driven exploration for mapless navigation with deep reinforcement learning." arXiv preprint arXiv:1804.00456 (2018).
- Liang, Jingsong, et al. "Context-aware deep reinforcement learning for autonomous robotic navigation in unknown area." Conference on Robot Learning. PMLR, 2023.
- Sathyamoorthy, Adarsh Jagan, et al. "Terrapn: Unstructured terrain navigation using online self-supervised learning." 2022 IEEE/RSJ International Conference on Intelligent Robots and Systems (IROS). IEEE, 2022.

*Limited discussion on exploration strategies*

The “minimum mapping ratio” metric is novel, but it’s not clear how exploration is coordinated globally, whether the Hamiltonian field inherently guides global coverage, or if it only reacts locally to perceived gradients.

**Questions:**

How does GRL-SNAM maintain global goal consistency when only local energy gradients are available? Is there a mechanism analogous to global value propagation (e.g., a learned potential update or memory-based field fusion)?

The learning process seems to optimize parameters of the Hamiltonian rather than a standard RL policy. Could the authors clarify whether the reward signal still drives policy gradient updates, or if this is a purely energy-based optimization (akin to unsupervised RL or active inference)?

How does the system behave in environments with dynamically changing obstacles or shifting goals? Does the energy-based formulation adapt online, or does it require retraining?

Could the authors compare with DRL-based navigation baselines (e.g., model-based/model-free, hierarchical RL, waypoint-/motion-based)? It would strengthen the claim that GRL-SNAM achieves similar or better coordination without an explicit hierarchy.

Given the Hamiltonian structure, could this framework naturally integrate with learned world models (e.g., latent-space dynamics) for true model-based RL on manifolds or navigation in 3D environments? It seems well-suited for that direction.

---

> ### Author Response · Authors · 2025-12-03
>
> **[W1] Unclear role of learning vs. hand‑crafted structure**
>
> We thank the reviewer for raising this point. Our framework is not performing hand‑crafted analytic control design; it is genuinely learning through interaction. Although the Hamiltonian has a structured form motivated by control theory, the learning process is fully data‑driven:
>
> * The agent collects trajectories under the current Hamiltonian field,
> * Updates the Hamiltonian parameters using observed rewards and adjoint feedback,
> * And then generates new trajectories under the updated field.
>
> This closed‑loop data generation is analogous to the policy/value iteration cycle in RL: the Hamiltonian plays the role of a learned energy landscape whose gradients define the policy, and the policy in turn refines the Hamiltonian. Thus, the method is far from fitting i.i.d. data; it is an interactive RL process where the dynamics of learning and data collection mutually reinforce each other. The Hamiltonian arises naturally from optimal control duality, but its parameters and behavior are learned, not hand‑specified.
>
> **[W2] Insufficient analysis of global consistency**
> We appreciate the reviewer’s concern regarding global navigation. Hamiltonian control provides a crucial property here: the Hamiltonian value is a local encoding of a long‑horizon objective. Because the optimality conditions in continuous‑time control reduce a global integral objective into local pointwise constraints (the Hamiltonian equations), global consistency is enforced implicitly by the dynamics. In practice, this yields globally consistent trajectories, loop‑closure without a map and long‑horizon goal pursuit, because every local gradient step is derived from a quantity that encodes the global objective.
>
> **[W3] Limited comparison to DRL baselines**
>
> We thank the reviewer for the suggestion. We have added comparisons with standard model‑free deep RL algorithms, PPO, TRPO, and SAC, using identical observation spaces, action parametrizations, and neural encoders to ensure fairness. The results show strong advantages in success rate, control accuracy, and sample efficiency:
>
> | Method    |   SPL ↑ | Detour ↓ | Min. Clearance (m) ↑ | Mapping Ratio (%) ↓ |
> |-----------|--------:|---------:|---------------------:|--------------------:|
> | PF        |     0.77  |    1.42   |            0.18   |           10.3   |
> | CBF       |     0.96  |     1.04   |           0.32   |           11.2   |
> | GRL-SNAM  |     0.95  |     1.09   |          0.26    |           10.7   |
> | SAC (SB3) |     0.07     |    1.65   |          -0.1 |            14.7   |
> | PPO (SB3) |     0.57  |     1.44   |            0.004  |           15.3  |
> | TRPO (SB3)|     0.57  |     1.53   |           0.004  |                14.6  |
>
> **[W4] Limited Discussion on Exploration Strategies**
> We thank the reviewer for this valuable question. Although our method is local in execution (following the Hamiltonian gradient), the energy field itself is globally conditioned on the entire obstacle configuration and goal location. As a result, each new environment induces a global reshaping of the Hamiltonian landscape, requiring the agent to react in a globally informed way.

---

> ### Author Response · Authors · 2025-12-03
>
> **[Q1–Q2]** These concerns are addressed in [W1] and [W2].
>
> **[Q3] Adaptation to dynamic obstacles or shifting goals**
>
> Our formulation includes an environment parameter $\mathcal{E}$ within the Hamiltonian, allowing online adaptation to dynamic obstacles and time‑varying goals. Because the agent learns generalizable obstacle‑interaction physics, new configurations produce immediate behavioral adjustments without retraining. We have revised our manuscript (Section 3.1-3.3) to reflect these justifications.
>
> **[Q4] Comparison with DRL navigation baselines**. These results are now included in [W3].
>
> **[Q5] Integration with learned world models**
> We agree this is a natural extension. The Hamiltonian structure interfaces cleanly with learned latent‑space dynamics, and combining our dual formulation with differentiable world models for 3D navigation is a promising direction we plan to pursue in future work.
> **[Q5] Integration with learned world models and 3D navigation**
>
> We agree that our Hamiltonian formulation is well-suited to integration with learned world models and navigation in 3D environments. Conceptually, the Hamiltonian operates on a state representation (physical or latent) and produces local generalized forces; a learned world model can supply that state by predicting future occupancy, obstacles, or manifold-valued latent dynamics. In such a setup, GRL-SNAM would act as a structure-preserving controller on top of a latent-space dynamics model, yielding a genuinely model-based Hamiltonian RL pipeline.
>
> We have already taken a first step in this direction by experimenting beyond simple 2D toy worlds. We evaluated GRL-SNAM on the indoor “DungeonMap’’ dataset from the context-aware navigation benchmark of Liang et al.[1], which exhibits many properties of realistic indoor navigation—narrow corridors, bottlenecks, long loops, and partial observability. In this setting, our method maintains robust performance under partial observability and long horizons:
>
> | Algorithm | Success Rate (\%) | Mean State Error (m) | Mean Goal Distance (m) | Convergence Steps        |
> |-----------|-------------------|----------------------|------------------------|--------------------------|
> | PPO       | 26.1              | 1.8                  | 1.2                    | 3.2M                     |
> | TRPO      | 21.7              | 2.1                  | 1.5                    | 3.8M                     |
> | SAC       | 18.4              | 2.4                  | 1.9                    | 4.1M                     |
> | GRL-SNAM  | **87.5**          | **0.3**              | **0.1**                | 500k (gradient steps)    |
>
> We are extending our pipeline to 3D indoor navigation in Habitat-Lab, with an initial focus on the HSSD-200 dataset[2]. A full 3D instantiation on HSSD-200 entails substantially longer training runs and nontrivial system integration (RGB-D sensing, top-down mapping, and robot-specific actuation). While we have a working prototype, we have not yet completed training and validation to a standard we consider suitable for reporting within the rebuttal window and available compute budget. We are continuing these experiments and will include the full 3D results, protocol, and metrics in the revised manuscript.
>
> **References**
> * [1] Liang, Jingsong, et al. "Context-aware deep reinforcement learning for autonomous robotic navigation in unknown area." Conference on Robot Learning. PMLR, 2023.
> * [2] Khanna, Mukul, et al. "Habitat synthetic scenes dataset (hssd-200): An analysis of 3d scene scale and realism tradeoffs for objectgoal navigation." CVPR, 2024.

---

### Official Review · Reviewer_8Pzq · 2025-11-01

**Soundness:** 2
**Presentation:** 2
**Contribution:** 2
**Rating:** 2
**Confidence:** 4

**Summary:**

The paper presents GRL-SNAM, a framework claiming to move beyond reinforcement learning (RL) by formulating navigation and mapping as Hamiltonian energy minimization on manifolds. It introduces “Differential Policy Optimization” (DPO) and claims that navigation arises as a gradient flow over learned Hamiltonians, evaluated on 2-D deformable-robot navigation tasks.

**Strengths:**

1. Physically grounded formulation:
Connects reinforcement learning with Hamiltonian optimal control, offering a structured and interpretable energy-based framework.


2. Geometric and multi-scale design:
Preserves energy and manifold structure through symplectic updates and coordinates multiple policies across different temporal scales.


3. Applicability to deformable robots:
Demonstrates potential in continuous and deformable robotic control, extending beyond traditional rigid-body navigation methods.

**Weaknesses:**

1. Theoretical Contributions Are Mischaracterized

Although the paper repeatedly claims to go beyond RL or beyond Bellman optimization, the theoretical formulation remains within standard optimal control and RL principles:
The proposed Hamiltonian is written as:

$
H(q, p) = K(p) + P(q)
$

which directly corresponds to the Pontryagin Hamiltonian in optimal control with a linear dynamic system without drift.
Its gradients are:

$$
\nabla_p H = f(x, u), \quad \nabla_q H = \nabla_x V.
$$

is equivalent to the value-function gradient in the Hamilton–Jacobi–Bellman (HJB) framework.

Thus, the “Hamiltonian gradient” proposed here produces the same optimal policy direction as the value gradient in standard optimal control and RL.

Claims that this method is “beyond Bellman” or “feedforward” are conceptually misleading — it effectively re-expresses the value-function gradient field under a different parameterization.

Hence, the theoretical novelty is overstated, and the method is a rebranding of existing value-based interpretations of RL/HJB rather than a fundamentally new control principle.

2. Experiments Are Limited and Not Convincing

The experiments focus exclusively on simple 2-D environments with “a hyperelastic ring navigating among several circular obstacles”
While the paper reports high success rates, these setups are not challenging:

Navigation among a few separated obstacles is a standard and well-solved problem; methods like [1] (and even classical potential field methods) already achieve similar results.

The authors compare only against simple baselines (PF, CBF, A*, DWA) but not against modern RL or geometric planning baselines in complex environments.

To substantiate the claimed scalability, the authors should evaluate on realistic indoor navigation benchmarks such as Habitat, Gibson, or Matterport3D, where partial observability, long horizons, and mapping uncertainty actually challenge RL and control approaches.

Without such experiments, the empirical section does not demonstrate generalization or scalability.

[1] Optimal obstacle avoidance based on the Hamilton–Jacobi–Bellman equation

**Questions:**

Several notational and structural issues hinder readability:

Key symbols in the reward/energy equations (Eq. 2) are undefined, including functions epsilon() and b()

---

> ### Author Response · Authors · 2025-12-03
>
> **[W1] Theoretical Contributions Are Mischaracterized**
>
> >Thus, the "Hamiltonian gradient" proposed here produces the same optimal policy direction as the value gradient in standard optimal control and RL.
>
> We appreciate the reviewer's insight that, under optimality, the solution produced by Hamiltonian gradient direction aligns that by the value-function gradient. This conceptual equivalence is indeed part of our motivation: since the dual and primal formulations yield the same optimal solution, it is natural to estimate the solution by learning from the dual (adjoint) dynamics.
>
> However, we clarify that our method is not a reparameterization of value-function gradients. In fact, no such functional reparameterization exists. The reviewer’s expression implicitly assumes $\nabla_q H = \nabla_x V$, but this does not hold in general. The closest valid correspondence is that the optimal adjoint variable $p^*(t)$ equals $\nabla_x V$, and thus $\nabla_q H$ has no equivalent interpretation in standard value-based methods.
>
> More importantly, this equivalence only holds along the optimal path and under ideal conditions. During learning, when policies and trajectories are far from optimal, adjoint variables and value gradients diverge significantly. Our approach is explicitly designed to operate in this regime: it learns adjoint dynamics directly, using a model-free, feedforward update that avoids value-function estimation or bootstrapping altogether.
>
> > Although the paper repeatedly claims to go beyond RL or beyond Bellman optimization...
>
> We apologize for any confusion caused by this phrasing. By "beyond Bellman", we mean going beyond the dynamic programming principle (DPP) that underlies nearly all value-based RL methods, including PPO, SAC, and TRPO. These methods are deeply tied to Bellman recursion and typically do not incorporate HJB equations or dual control theory.
>
> In contrast, our work follows a small but growing line of model-free methods that depart from DPP, including:
> * Settai et al. (2025): Model-free HJB-based learning.
> * Jia & Zhou (2023): Martingale formulation of DPP.
> * Nguyen & Bajaj (2025): Duality-based Hamiltonian RL, which our work builds upon and extends.
>
> We hope this clarifies that our approach is not a re-expression of value-function gradients, but a distinct dual formulation, grounded in Hamiltonian dynamics.

---

> ### Author Response · Authors · 2025-12-03
>
> **[W2] Experiments Are Limited and Not Convincing**
> We appreciate the reviewer's concerns and agree that demonstrating scalability and generalization is essential. We address this in three parts:
>
> 1. We have now included benchmarks against standard model-free RL methods: PPO, TRPO, and SAC, using identical observation spaces, action representations, and feature encoding architectures to ensure fair comparison. Our results below show that GRL-SNAM significantly outperforms these methods in success rate, control accuracy, and mapping ratio:
>
> | Method    |   SPL ↑ | Detour ↓ | Min. Clearance (m) ↑ | Mapping Ratio (%) ↓ |
> |-----------|--------:|---------:|---------------------:|--------------------:|
> | PF        |     0.77  |    1.42   |            0.18   |           10.3   |
> | CBF       |     0.96  |     1.04   |           0.32   |           11.2   |
> | GRL-SNAM  |     0.95  |     1.09   |          0.26    |           10.7   |
> | SAC (SB3) |     0.07     |    1.65   |          -0.1 |            14.7   |
> | PPO (SB3) |     0.57  |     1.44   |            0.004  |           15.3  |
> | TRPO (SB3)|     0.57  |     1.53   |           0.004  |                14.6  |
>
>
>
> 2. 2D DungeonMap and 3D Indoor Navigation (Habitat / HSSD-200)
> To further demonstrate generalization beyond simple 2D layouts, we have conducted additional experiments on the indoor “DungeonMap’’ dataset from the context-aware navigation benchmark of Liang et al.~[4]. This dataset already exhibits many properties of realistic indoor navigation—narrow corridors, bottlenecks, long loops, and partial observability. In this setting, our method continues to perform robustly under partial observability and long horizons:
>
> | Algorithm | Success Rate (%) | Mean State Error (m) | Mean Goal Distance (m) | Convergence Steps |
> |-----------|-----------------|-------------------|----------------------|------------------|
> | PPO      | 26.1           | 1.8              | 1.2                 | 3.2M             |
> | TRPO     | 21.7           | 2.1              | 1.5                 | 3.8M             |
> | SAC      | 18.4           | 2.4              | 1.9                 | 4.1M             |
> | GRL-SNAM | **87.5**          | **0.3**              | **0.1**                 | 500k (gradient steps) |
>
> In parallel, we have begun extending our pipeline to 3D indoor navigation benchmarks in Habitat-lab, focusing on the HSSD-200 dataset~[5]. A full 3D pipeline on HSSD-200 requires substantially longer training runs and careful system integration (RGB-D sensing, top-down mapping, and robot-specific actuation), which we could not complete, validate, and cleanly report within the rebuttal timeframe and available GPU budget. We are continuing these experiments and will include the full 3D results, protocol, and metrics in the revised manuscript.
> We emphasize that the DungeonMap results already demonstrate that GRL-SNAM scales to complex, indoor-style layouts with narrow passages and partial observability, and that its local-force formulation naturally extends to top-down navigation layers in 3D environments, with only an additional policy interface needed to map GRL-SNAM’s continuous commands to discrete robot actions.
>
> 3. We thank the reviewer for pointing out [1]. However, the referenced method is model-based and tailored to a constant unit-reward setting, whereas our work addresses goal-conditioned reward structures and dynamically changing environments. This difference in problem formulation leads to fundamentally different algorithmic choices and practical behavior.
>
> **[Q1-Q2] Symbol, Notation and Structure Improvement.**
>
> We have revised Section 3 in our manuscript to improve the presentation of the GRL-SNAM algorithm and to clarify the notation originally introduced in Eq. (2).
>
> **References**
> * [1] Settai, H., Takeishi, N., & Yairi, T. A Temporal Difference Method for Stochastic Continuous Dynamics. arXiv preprint arXiv:2505.15544, 2025.
> * [2] Nguyen, M., & Bajaj, C. A Differential and Pointwise Control Approach to Reinforcement Learning. arXiv preprint 2404.15617, 2025.
> * [3] Jia, Z., & Zhou, X. Y. q-Learning in Continuous Time. Journal of Machine Learning Research, 24(178):1–58, 2023.
> * [4] Liang, Jingsong, et al. "Context-aware deep reinforcement learning for autonomous robotic navigation in unknown area." Conference on Robot Learning. PMLR, 2023.
> * [5] Khanna, Mukul, et al. "Habitat synthetic scenes dataset (hssd-200): An analysis of 3d scene scale and realism tradeoffs for objectgoal navigation." Proceedings of the IEEE/CVF Conference on Computer Vision and Pattern Recognition. 2024.

---

### Meta-Review · Area_Chair_N9Ys · 2026-01-06

**Summary:**

Reviewer 8Pzq
- "Theoretical Contributions Are Mischaracterized" / the theoretical novelty is overstated
- "Experiments Are Limited and Not Convincing"; "To substantiate the claimed scalability, the authors should evaluate on realistic indoor navigation benchmarks such as Habitat, Gibson, or Matterport3D, where partial observability, long horizons, and mapping uncertainty actually challenge RL and control approaches."

Reviewer eoAa
- Unclear role of learning vs. hand-crafted structure
- Insufficient analysis of global consistency
- Limited comparison to DRL baselines
- Limited discussion on exploration strategies

Reviewer aWJS
- experimental validation is limited
- frequent jumps in the presentation

**Reviewer Concerns:**

Addressed concerns:
All except the one below.

Partially addressed concerns:

Reviewer 8Pzq
- "Theoretical Contributions Are Mischaracterized" / the theoretical novelty is overstated.
The authors rebutted with some justification of the theoretical novelty in response to this concern. Their justification is plausible, although fairly terse and not quite rigorous. Whether this justification can be expanded to be clearer and sufficiently rigorous, is unclear to me, so it's plausible that Reviewer 8Pzq would have either accepted or dismissed this explanation.

**Reviewer Scores:**

- Reviewer 8Pzq: 2->4 or 6
- Reviewer eoAa: 6->6 or 8
- Reviewer aWJS: 6->6 or 8

---

### Decision · Program_Chairs · 2026-01-26

Accept (Poster)